# NoisyCoconut: Counterfactual Consensus via Latent Space Reasoning

**Michael Jerge**                                                    *mj6ux@virginia.edu*
*Department of Computer Science University of Virginia*

**David Evans**                                                      *evans@virginia.edu*
*Department of Computer Science University of Virginia*

## Abstract

This paper presents NOISYCOCONUT, a novel inference-time method that enhances large language model (LLM) reliability using multiple executions with manipulated internal representations. Unlike fine-tuning methods that require extensive retraining, NOISYCOCONUT operates directly on model representations during inference and requires no retraining. Rather than training models to reason in latent space, we inject controlled noise into latent trajectories to generate diverse reasoning paths. Agreement among these paths provides a confidence signal, enabling models to abstain when uncertain. We demonstrate that this approach achieves effective coverage-accuracy tradeoffs across multiple reasoning benchmarks, providing a practical pathway to improving the reliability of LLM outputs without requiring any retraining. Our experiments show that unanimous agreement among noise-perturbed paths reduces error rates from 40–70% to below 15%, enabling models to exceed 95% accuracy on mathematical reasoning tasks through selective abstention. Our code is available at `https://github.com/mmjerge/noisycoconut`.

## 1 Introduction

Large language models (LLMs) continue to demonstrate remarkable capabilities and are increasingly deployed in high-stakes environments, including healthcare, financial services, and legal practice (Haltaufderheide & Ranisch, 2024). These nondeterministic models generate text autoregressively, producing one token at a time conditioned on the preceding token sequence based on a probability distribution. While this process yields fluent and coherent outputs, it provides no inherent mechanism for the model to signal uncertainty about its own generations. As a result, LLMs can produce seemingly confident but erroneous outputs, a phenomenon known as hallucination (Xu et al., 2025b; Farquhar et al., 2024). Such errors pose particular risks in regulated sectors, where legal and regulatory obligations demand accuracy and auditability.

One approach to mitigate the risk of unreliable outputs in predictive machine learning systems is selective prediction, which allows models to abstain from giving predictions when it is likely to be incorrect (Geifman & El-Yaniv, 2017; El-Yaniv & Wiener, 2010). Selective prediction introduces a tradeoff between *coverage* (the fraction of queries answered) and *accuracy* (the correctness of answered queries). Selective prediction is well-established in classification settings where softmax confidence provides a natural abstention signal (Geifman & El-Yaniv, 2017; Guo et al., 2017). Enabling confidence-based abstention for generative LLMs is less straightforward, however, because token-level probabilities often reflect linguistic uncertainty rather than factual correctness (Kuhn et al., 2023). Selective prediction requires a confidence signal that is well correlated with the prediction accuracy, but such a signal is not directly available in LLMs.

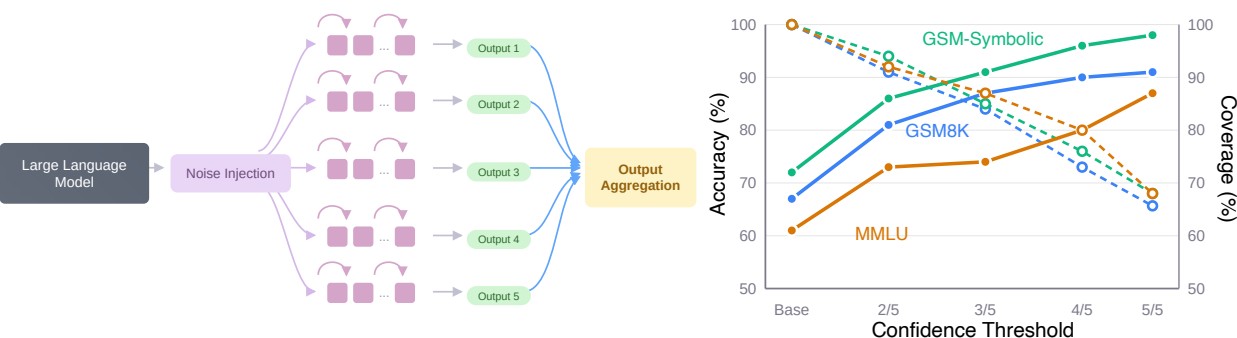

Figure 1: (Left) NOISYCOCONUT architecture with noise-induced branching for diverse reasoning paths. (Right) Accuracy-coverage tradeoff for Qwen2.5-7B-Instruct across three benchmarks. Solid lines show accuracy, while the dashed lines show coverage.

Several methods have been proposed for estimating the confidence of generative LLMs at inference-time. Verbalized confidence methods prompt models to explicitly rate their certainty, though such self-assessments can be poorly calibrated (Kadavath et al., 2022; Xiong et al., 2024). Self-correction approaches (e.g., Madaan et al., 2023; Shinn et al., 2023) iteratively refine outputs through self-generated feedback, though recent work suggests that intrinsic self-correction struggles for reasoning tasks without external signals (Huang et al., 2024). Sampling-based methods generate multiple outputs and measure their consistency, using majority voting across multiple, generated paths or clustering semantically equivalent responses to compute uncertainty over meanings (e.g., Wang et al., 2023; Kuhn et al., 2023). When sampled outputs disagree, the system can abstain, enabling coverage–accuracy tradeoffs. However, because these methods rely on standard autoregressive sampling, they are restricted to the diversity present in the surface-level token distribution at the output layer (Holtzman et al., 2020). Consequently, they often fail to explore a sufficiently broad set of reasoning trajectories, as the sampling process is confined by the language space.

A separate line of research has begun exploring methods that operate directly on the continuous hidden states of language models, rather than through token generation (Zhu et al., 2025b; Goyal et al., 2023; Hao et al., 2025). These approaches reflect a growing recognition that the discrete, token-by-token generation process may be suboptimal for complex reasoning tasks and that reasoning in neural networks involves operations in a continuous representational space not readily accessible through token manipulations. However, latent-space methods face the same reliability challenges as their token-space counterparts, as models can produce incorrect outputs with no inherent mechanism for assessing confidence. Moreover, all of these methods require specialized architectures and training to enable latent-space reasoning.

We address this gap by extending agreement-based confidence estimation to the latent space. We build on the Continuous Chain-of-Thought (COCONUT) framework from Hao et al. (2025), which trains language models to perform iterative computation in hidden states by feeding the last hidden state back as the next input embedding. Our goal differs from COCONUT's—rather than training models to reason in latent space, we leverage latent representations to assess confidence in model outputs. Specifically, we introduce NOISYCOCONUT, illustrated in Figure 1. NOISYCOCONUT injects controlled noise into latent trajectories to generate counterfactual reasoning paths. This enables selective prediction for latent-space methods, achieving effective coverage–accuracy tradeoffs without additional training.

**Contributions.** We introduce NOISYCOCONUT, a training-free inference-time method that generates diverse reasoning paths through controlled noise injection in continuous latent space, enabling agreement-based confidence estimation directly on hidden states (Section 3). Then, we demonstrate that path agreement provides a reliable proxy for prediction confidence (Section 4). Section 4.2 characterizes the noise-accuracy relationship in latent space. We show empirically that NOISYCOCONUT achieves effective coverage-accuracy tradeoffs across six reasoning benchmarks and six language models (Section 4.3).

## 2 Background and Related Work

Our work connects two research threads: inference-time methods for improving LLM reliability and latent-space computation. We review both areas next, and describe the CoCONUT method, which our work builds upon directly in Section 2.3.

### 2.1 Confidence Estimation and Selective Prediction

Selective prediction for LLMs in generative settings presents fundamentally different challenges than selective prediction for classification tasks (Xin et al., 2021). In classification, models select from a fixed set of discrete labels, but autoregressive generation produces outputs token-by-token over many steps, with an effectively unbounded space of possible responses (Ren et al., 2023). This distinction has motivated a substantial body of work developing uncertainty quantification methods tailored for generative models (Si et al., 2023; Varshney et al., 2022; Kadavath et al., 2022; Kuhn et al., 2023). Prior works have examined selective prediction in settings where questions themselves are ambiguous, requiring models to recognize when clarification is needed rather than producing potentially incorrect answers (Cole et al., 2023; Yin et al., 2023). Another line of research employs auxiliary models trained to discriminate between correct and incorrect predictions from a primary QA system (Varshney & Baral, 2023), though such approaches introduce additional training requirements and may not generalize across domains.

These challenges have motivated sampling-based approaches that operationalize selective prediction by generating multiple outputs and using their agreement as a confidence signal for abstention decisions. Self-consistency (Wang et al., 2023) samples $N$ reasoning chains from the model, then selects the most frequent final answer via majority voting. The key insight is that correct answers tend to be reachable via multiple reasoning paths, while incorrect answers arise from idiosyncratic errors unlikely to recur across samples. This approach substantially improves accuracy on arithmetic and commonsense reasoning benchmarks, though it requires generating $N$ complete token sequences per query.

An alternative direction uses training to improve selective prediction. ASPIRE (Chen et al., 2023b) fine-tunes LLMs via parameter-efficient tuning to not only answer questions but also evaluate whether their generated answers are correct, producing explicit confidence scores. While effective, such approaches require task-specific training data and model access, limiting applicability.

Semantic entropy (Kuhn et al., 2023; Farquhar et al., 2024) addresses a limitation of token-level uncertainty measures since different surface forms can express the similar meanings. Rather than computing entropy over token sequences, semantic entropy clusters sampled responses by meaning using bidirectional entailment, then computes entropy over these semantic equivalence classes. This provides more robust uncertainty estimates that have been shown to detect hallucinations more reliably than token probabilities alone.

Verbalized confidence methods take a different approach, prompting models to directly express uncertainty. Kadavath et al. (2022) introduced $P(\text{True})$, which asks models to evaluate whether their own sampled answers are correct; the probability assigned to the token "True" serves as a confidence estimate. While effective for well-calibrated models, subsequent work has shown that verbalized confidence can be poorly calibrated, particularly for instruction-tuned models (Xiong et al., 2024; Tian et al., 2023). Probing-based methods instead train classifiers on internal representations to predict correctness (Azaria & Mitchell, 2023), though these require labeled data for each task.

Recent work has extended these approaches in various directions. Confidence-Informed Self-Consistency (Taubenfeld et al., 2025) weights the majority vote by model confidence, reducing the number of samples needed. Kernel Language Entropy (Nikitin et al., 2025) generalizes semantic entropy using kernel methods for finer-grained uncertainty estimates.

### 2.2 Inference-Time Approaches to Improving LLM Reliability

Beyond confidence estimation, several inference-time methods seek to improve LLM reliability by structuring the generation process itself. These methods can operate fully in the token space without any consideration

of internal model states, or can take advantage of the opportunity to observe and manipulate internal representations. We briefly review both types of work next.

**Token-Space Methods.** Chain-of-Thought (Wei et al., 2022) improves accuracy on complex tasks by eliciting intermediate steps, reducing errors that arise from single-step generation. This approach has been extended through decomposition strategies (Khot et al., 2022; Zhou et al., 2022) and explicit search methods such as Tree-of-Thoughts (Yao et al., 2023), which explores multiple solution paths and enables backtracking. Jerge & Evans (2025) provides a more comprehensive survey of these methods.

Self-correction approaches attempt to improve reliability through iterative refinement. Self-Refine (Madaan et al., 2023) prompts models to critique and revise their own outputs, while Reflexion (Shinn et al., 2023) maintains memory of past errors to guide future attempts. However, Huang et al. (2024) demonstrate that intrinsic self-correction, without external feedback, often fails to improve accuracy on reasoning tasks.

These methods operate entirely in the language and token-space, generating explicit text at each step, limiting intermediate computation. As Madaan & Yazdanbakhsh (2022) observe, most tokens in a reasoning chain serve textual coherence rather than substantive computation.

**Latent-Space Methods.** An alternative approach directly manipulates the hidden states of language models, bypassing token generation. Studies have shown that intermediate reasoning steps can be recovered from hidden representations (Yang et al., 2024) and that models may employ latent processes distinct from their generated text (Turpin et al., 2024). Zhu et al. (2025b) categorize latent-space methods into activation-based approaches that expand computational depth through repeated layer processing, hidden state-based methods that maintain compressed memory states, and diffusion-based approaches enabling iterative refinement. Prior work has also explored augmenting LLMs with special tokens that enable additional hidden-state computation, such as learnable pause tokens (Goyal et al., 2023) and filler tokens (Pfau et al., 2024), though these require training and still operate through the token interface.

## 2.3 Continuous Chain-of-Thought

Our method builds directly on the Continuous Chain-of-Thought (Coconut) framework of Hao et al. (2025).

Standard autoregressive generation produces one discrete token at a time, forcing all intermediate computation to pass through the token interface. Coconut instead enables the model to reason in a continuous latent space. Rather than decoding an intermediate token and re-embedding it, Coconut feeds the model's last hidden state back as the next input embedding. The model enters this latent mode at a special *begin-of-thought* token (`<bot>`) and exits at an *end-of-thought* token (`<eot>`), after which it resumes ordinary token generation to produce the final answer.

Let $\mathcal{M}$ be a pre-trained language model with hidden dimension $d$, and let $f_\theta : \mathbb{R}^d \to \mathbb{R}^d$ denote a single forward pass through the transformer that maps a current hidden state to the next one. For an input query $q \in \mathcal{Q}$, let $\mathbf{h}_0 = \phi(q) \in \mathbb{R}^d$ denote the initial hidden state produced upon entering latent mode.

Coconut performs $T$ latent reasoning steps via the iteration

$$\mathbf{h}_{t+1} = f_\theta(\mathbf{h}_t), \tag{1}$$

after which the final hidden state $\mathbf{h}_T$ is decoded back into the language space to yield an output $y$. This formulation enables reasoning without intermediate tokenization and achieves strong performance on tasks requiring search and planning.

## 3 NoisyCoconut

We emphasize at the outset that NoisyCoconut does not train models to reason better in latent space, nor does it modify Coconut's learned dynamics. Rather, NoisyCoconut is a training-free *latent-space ensemble* for uncertainty estimation. It perturbs a pre-trained model's latent trajectory to generate an ensemble of reasoning paths, using their agreement as a confidence signal for selective prediction. In this sense

NOISYCOCONUT is the latent-space analogue of self-consistency (Wang et al., 2023), where self-consistency samples diverse token-level reasoning chains, NOISYCOCONUT induces diversity directly in the continuous hidden state.

## 3.1 Noise-Based Path Exploration

The core idea is to perturb the hidden states during the continuous latent-state process, creating multiple reasoning trajectories. We sample independent Gaussian noise and inject it into the latent trajectory to create $K$ reasoning paths from a common initial state. The noise should be large enough to induce meaningful divergence, yet constrained enough to maintain coherent sequences. If the resulting paths reach the same conclusion, this increases our confidence that the prediction is correct; if their agreement is below a set confidence threshold, we abstain.

Using the notation from Section 2.3, the NOISYCOCONUT process evolves as

$$\mathbf{h}_{t+1} = f_\theta(\mathbf{h}_t + \boldsymbol{\eta}_t), \quad \boldsymbol{\eta}_t \sim \mathcal{N}(\mathbf{0}, \sigma_t^2 \mathbf{I}_d), \tag{2}$$

where the $\{\boldsymbol{\eta}_t\}$ are independent and the noise scale decays exponentially:

$$\sigma_t = \sigma_0 e^{-\lambda t}, \quad \sigma_0, \lambda > 0. \tag{3}$$

Compared with the unperturbed COCONUT update in Equation 1, the only change is the additive perturbation $\boldsymbol{\eta}_t$. Setting $\sigma_0 = 0$ recovers COCONUT exactly.

We adapt the noise to the trajectory's scale so that perturbations remain proportionate to the magnitude of the hidden state. Let $\mu_t$ denote the exponentially-weighted moving average of the hidden state norm,

$$\mu_t = \alpha \mu_{t-1} + (1 - \alpha)\|\mathbf{h}_t\|_2, \quad \alpha \in (0, 1), \tag{4}$$

giving the adaptive noise scale

$$\sigma_t = \sigma_0 e^{-\lambda t} \cdot \frac{\|\mathbf{h}_t\|_2}{\mu_t}. \tag{5}$$

## 3.2 Path Diversity

Effective exploration requires trajectories that are sufficiently distinct rather than minor variations of a single solution strategy. For $K$ paths $\{\mathbf{h}_{0:T}^{(i)}\}_{i=1}^K$, we define pairwise trajectory diversity as

$$\mathcal{D}_K = \frac{2}{K(K-1)} \sum_{1 \le i < j \le K} \frac{1}{T} \sum_{t=0}^{T-1} \|\mathbf{h}_t^{(i)} - \mathbf{h}_t^{(j)}\|_2. \tag{6}$$

Under mild non-degeneracy assumptions on the latent dynamics, the expected diversity scales with the injected noise, $\mathbb{E}[\mathcal{D}_K] = \Omega(\sigma_0 \sqrt{dT})$, indicating that $\sigma_0$ provides a direct control over ensemble spread. We give a first-order derivation in Appendix A and confirm the scaling empirically (see Table 2 in Section 4.2).

## 3.3 Aggregating Outputs

For our setting we assume a finite output set $\mathcal{Y}$. Each of the $K$ paths yields an output $y^{(i)} \in \mathcal{Y}$. We aggregate these into an output result $\hat{y}$, which is either a consensus prediction or abstention (denoted $\perp$). For our experiments, we aggregate by voting, selecting the most frequent output and abstaining when no answer exceeds that threshold:

$$\hat{y} = \begin{cases} \hat{y}^\star & \text{if } \exists \hat{y}^\star \in \mathcal{Y} \text{ s.t. } \sum_{i=1}^K \mathbb{I}\{y^{(i)} = \hat{y}^\star\} > \tau \\ \perp & \text{otherwise} \end{cases} \tag{7}$$

---

**Algorithm 1:** NoisyCoconut Inference

---

**Input:** query $q$; pre-trained language model $f_\theta$ (with encoder $\phi$); number of paths $K$; latent steps $T$;
 base noise scale $\sigma_0$; decay rate $\lambda$; agreement threshold $\tau$; noise schedule
 $\in \{\texttt{fixed}, \texttt{adaptive}, \texttt{none}\}$

**Output:** consensus answer $\hat{y}^\star$, or abstention $\perp$

---

**1**  $\mathbf{h}_0 \leftarrow \phi(q)$             *// initial hidden state at entry to latent mode*

**2**  **for** $k \leftarrow 1$ **to** $K$ **do**

  *// generate one perturbed latent trajectory*

**3**   $\mu \leftarrow \|\mathbf{h}_0\|_2$

**4**   $\mathbf{h} \leftarrow \mathbf{h}_0$

**5**   **for** $t \leftarrow 0$ **to** $T - 1$ **do**

**6**    **if** *schedule* = $\texttt{none}$ **then**

**7**     $\sigma_t \leftarrow 0$

**8**    **else if** *schedule* = $\texttt{adaptive}$ **then**

**9**     $\mu \leftarrow \alpha\,\mu + (1 - \alpha)\,\|\mathbf{h}\|_2$

**10**     $\sigma_t \leftarrow \sigma_0\,e^{-\lambda t} \cdot \|\mathbf{h}\|_2/\mu$        *// Equation 5*

**11**    **else**

**12**     $\sigma_t \leftarrow \sigma_0\,e^{-\lambda t}$          *// exponential decay*

**13**    $\boldsymbol{\eta}_t \sim \mathcal{N}(\mathbf{0}, \mathbf{I}_d)$

**14**    $\mathbf{h} \leftarrow f_\theta(\mathbf{h} + \sigma_t\,\boldsymbol{\eta}_t)$    *// perturb, then feed hidden state back as next input embedding*

**15**   $y^{(k)} \leftarrow \text{DECODE}(f_\theta, \mathbf{h})$       *// exit latent mode, greedy decoding*

 *// aggregate by majority vote (Equation 7)*

**16**  $V \leftarrow \{\!\{\, y^{(1)}, \ldots, y^{(K)} \,\}\!\}$         *// multiset of path answers*

**17**  $(\hat{y}^\star,\, c) \leftarrow$ most frequent answer in $V$ and its count

**18**  **if** $c \geq \tau$ **then**

**19**  **return** $\hat{y}^\star$           *// sufficient agreement: emit consensus*

**20**  **else**

**21**  **return** $\perp$           *// insufficient agreement: abstain*

---

The agreement count of the top answer serves as a tunable confidence threshold $\tau \geq K/2$: lowering $\tau$ increases coverage at the cost of accuracy, while raising it does the reverse, yielding the accuracy–coverage tradeoff that is the goal of selective prediction. For our experimental analysis ($K = 5$), we categorize the consensus patterns into mutually exclusive outcomes to diagnose reliability based on the minimum number of agreeing paths:

- **Unanimous (**$5/5$**):** all five paths converge to the same answer.

- **Strong Majority (**$4/5$**):** four paths agree, while one diverges.

- **Moderate Majority (**$3/5$**):** three paths agree.

- **Minimal Plurality (**$2/5$**):** two paths agree while the others are distinct (e.g., A, A, B, C, D).[1]

These categories map the latent stability of the model to the correctness of the final output, and correspond to thresholds $\tau \in \{5, 4, 3, 2\}$ in a deployed system. When no answer appears on more than two paths, the system abstains.[2]

The full method is summarized in Algorithm 1.

---

[1] Note for cases where $\tau < K/2$, Equation 7 would need to be more complex to express the constraint on the other paths.
[2] One could imagine future work that recovers an answer even without explicit agreement, or that exploits token probabilities or other path characteristics. For simplicity, here we keep aggregation simple by just counting outcomes.

Table 1: Language models used in evaluation

| Model | Parameters | Type |
|-------|-----------|------|
| Qwen2.5-7B-Instruct | 7B | Instruction-tuned |
| Qwen3-8B | 8B | Hybrid reasoning |
| Llama-3.1-8B-Instruct | 8B | Instruction-tuned |
| Mixtral-8x7B-Instruct-v0.1 | 8x7B | Instruction-tuned (MoE) |
| gpt-oss-20B | 20B | Foundation model |
| DeepSeek-R1-Distill-Qwen-7B | 7B | Distilled reasoning |

## 4  Evaluation and Results

To understand the coverage–accuracy tradeoffs enabled by NOISYCOCONUT, we conduct a systematic evaluation across six language models and six benchmarks. Section 4.1 describes our experimental setup, including model selection, benchmarks, and implementation details. We first validate a core assumption of our method by characterizing the noise-accuracy relationship as the noise scale varies (Section 4.2). Section 4.3 presents our main results, analyzing how agreement patterns among diverse reasoning paths correlate with accuracy. Our findings show that path agreement in latent space provides a strong signal for prediction reliability. Unanimous agreement among five noise-perturbed reasoning paths reduces error rates from 40–70% to below 15%, enabling models to achieve over 95% accuracy on mathematical reasoning benchmarks when selectively abstaining on low-confidence predictions.

### 4.1  Experimental Setup

We selected representative open-source language models of similar parameter scales to evaluate the effectiveness of NOISYCOCONUT across different architectures and training paradigms. The selection includes a mix of instruction-tuned models, base models, and a specialized distilled reasoning model to assess performance across different model optimization approaches.

Table 1 summarizes the models tested. Qwen2.5-7B-Instruct (Qwen Team, 2024) is an instruction-tuned model from the Qwen family, known for strong reasoning capabilities despite its moderate size. Qwen3-8B (Yang et al., 2025) is a more recent addition to the Qwen model suite, consisting of a hybrid architecture for reasoning across a range of downstream tasks. Llama-3.1-8B-Instruct (Grattafiori et al., 2024) is one of Meta's smaller, instruction-tuned variants in the Llama series, widely used as a foundation for many applications. Mixtral-8x7B-Instruct-v0.1 (Jiang et al., 2024) provides another instruction-tuned variant, optimized specifically for following complex instructions with enhanced reasoning capabilities. The gpt-oss-20B (OpenAI, 2025) model is an open-weights model that offers strong general-purpose language abilities. DeepSeek-R1-Distill-Qwen-7B (DeepSeek-AI, 2025) is a specialized distilled reasoning model that combines elements from both DeepSeek and Qwen architectures, where knowledge distillation techniques were specifically employed to enhance reasoning capabilities.

While our NOISYCOCONUT implementation works directly with standard architectures, we observed that gpt-oss-20B produced degenerate outputs when using the default final-layer hidden state feedback mechanism, regardless of noise level. Thus, we use $\mathbf{h}^{(1)}$ (first layer) instead of $\mathbf{h}^{(L)}$ (final layer) for the continuous sequence. This modification enables functional operation, though it reduces the effective reasoning depth per latent pass. All other models use the standard final-layer configuration.

**Benchmarks.** To evaluate both mathematical reasoning and broader knowledge capabilities, we selected six established benchmarks. GSM8K (Cobbe et al., 2021) provides a challenging grade-school math word problem benchmark consisting of 1000 linguistically diverse problems requiring multi-step reasoning. GSM-Symbolic (Mirzadeh et al., 2025) represents a variant of GSM8K that replaces natural language descriptions with more symbolic representations of the same underlying problems, testing how well models handle different formulations of equivalent mathematical concepts. The MATH dataset (Hendrycks et al., 2021b) consists of challenging competition mathematics problems spanning a range of subjects and difficulty levels, requiring

detailed multi-step solutions. The MMLU (Massive Multitask Language Understanding) (Hendrycks et al., 2021a) benchmark offers a comprehensive assessment covering 57 subjects across STEM, humanities, social sciences, and more, testing both factual knowledge and reasoning abilities. GPQA (Rein et al., 2024) is a graduate-level question-answering benchmark featuring multiple-choice questions in biology, physics, and chemistry. GPQA-Diamond (Rein et al., 2024) is a subset of GPQA, comprising of questions that experts answer correctly while non-experts consistently fail.

For the larger benchmarks (GSM8K, GSM-Symbolic, and MMLU), we randomly sampled 1000 questions each to ensure sufficient statistical power while maintaining computational feasibility. For MATH, we evaluated on the full 500-question MATH-500 test split, the standard evaluation subset of the original MATH benchmark Hendrycks et al. (2021b). For GPQA and GPQA-Diamond, we used all available questions (448 and 198, respectively), as these sets are smaller than our original target sample size.

## 4.2 Validating Impact of Noise

A fundamental assumption of NoisyCoconut is that injecting noise into hidden states produces controlled and predictable degradation in model performance. To validate this assumption and inform our choice of noise level, we systematically measured accuracy across noise scales $\sigma \in [0, 50]$, where $\sigma$ represents the ratio of noise norm to hidden state norm ($\|\boldsymbol{\epsilon}\|_2 = \sigma\|\mathbf{h}\|_2$).

Figure 2 summarizes the results, showing a characteristic sigmoid decay pattern across all models, with fits achieving $R^2 \geq 0.94$. Accuracy remains relatively stable at low noise levels before undergoing steep degradation, eventually plateauing near random performance. This consistent pattern across architectures confirms that noise injection produces predictable, exploitable effects.

Based on these results, we selected $\sigma_0 = 0.2$ (marked by the vertical dotted line in Figure 2) as the noise level for our main experiments. This value lies in the early portion of the degradation curve, where accuracy remains within 2–5% of unperturbed performance. This choice introduces sufficient perturbation to generate meaningful diversity in generated paths, while avoiding excessive degradation that would compromise individual path quality. At this noise level the perturbed paths are also clearly separated; we report the corresponding path-diversity measurements ($\mathcal{D}_K$), which grow monotonically with $\sigma_0$ as predicted by Section 3.2, together with their derivation in Appendix A. Further analysis of this trade-off is provided in Appendix C.

## 4.3 Results and Analysis

We implemented NoisyCoconut as described in Section 3, with several key hyperparameters selected to balance performance and computational cost.

For our main experiments, we use five reasoning paths per problem, which our ablation studies (Appendix C) show provides a good efficiency–performance tradeoff. Based on our experiments in Section 4.2 we set the noise scale to $\sigma_0 = 0.2$. Noise injections were performed at the first hidden state after the first forward pass. Early injection enables greater exploration of the reasoning space and aligns with our goal of generating diverse reasoning trajectories from the outset. We set a maximum of eight latent thinking steps to allow sufficient reasoning depth, following the same constraint established in the original Coconut work.[3]

For each model and benchmark combination, we first establish a baseline by evaluating the model using standard greedy decoding without inference-time modifications. We then apply Noisy Coconut, generating five ($K = 5$) distinct reasoning paths for each test example by injecting controlled noise into the hidden states. We analyze the generated paths using the aggregation outcomes defined in Section 3. While we track distinct outcomes such as "Split Votes" for error analysis, our primary performance evaluation focuses on the cumulative confidence threshold. We perform this generation step once per test example, and then analyze the impact of applying the different aggregation functions to this static set of outputs. This allows us to observe how the consensus output $\hat{y}$, which is either a specific answer $\hat{y}^\star$ or an abstention $\perp$, changes as a function of the strictness of the agreement threshold.

---

[3]Note that these component-wise ablations do not capture interaction effects between hyperparameters.

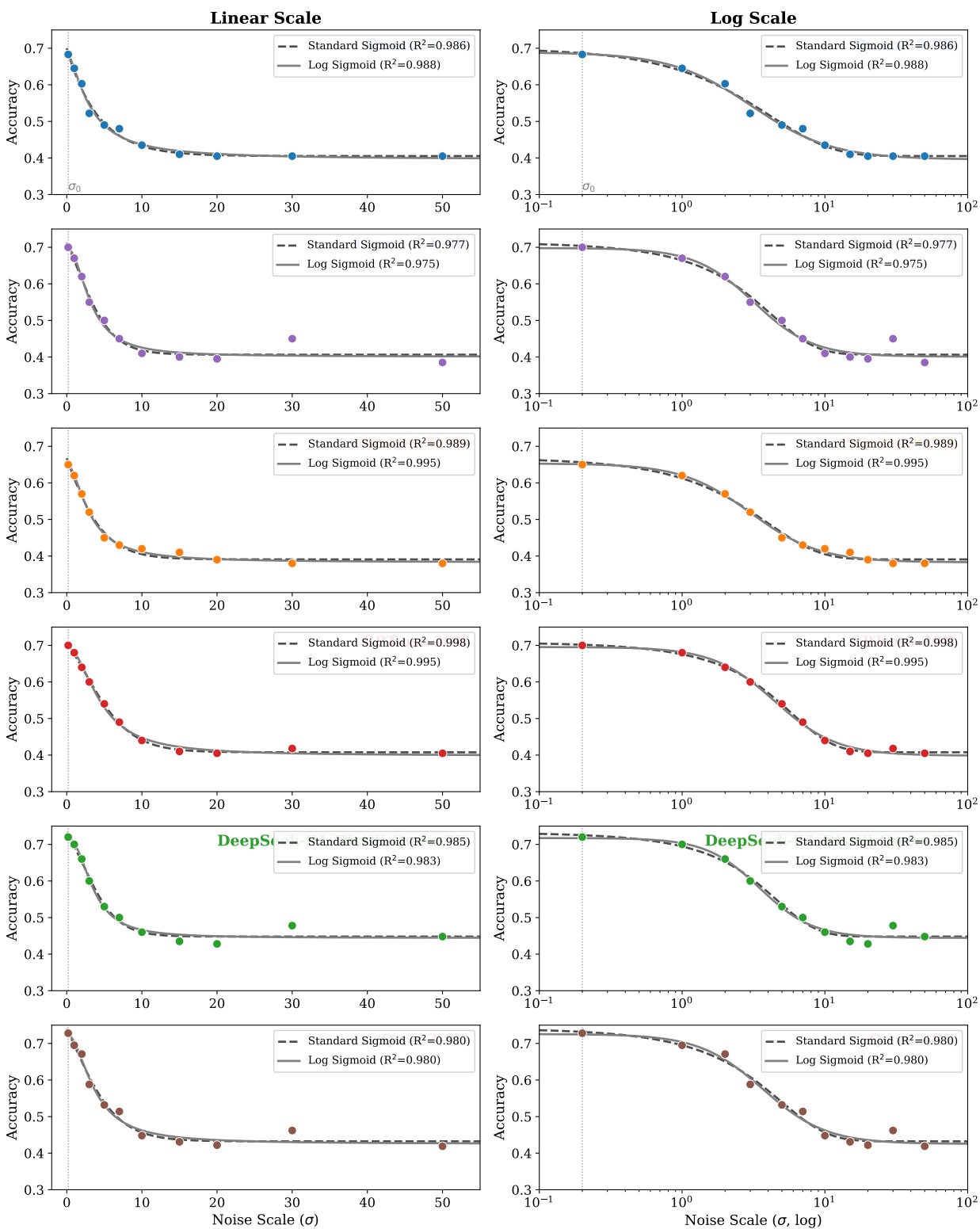

Figure 2: Accuracy degradation vs. noise scale $\sigma$ on linear (left) and logarithmic (right) scales. The vertical dotted line marks $\sigma_0 = 0.2$, chosen to balance perturbation strength with model performance (see Appendix C). Sigmoid fits achieve $R^2 \geq 0.94$ for all models.

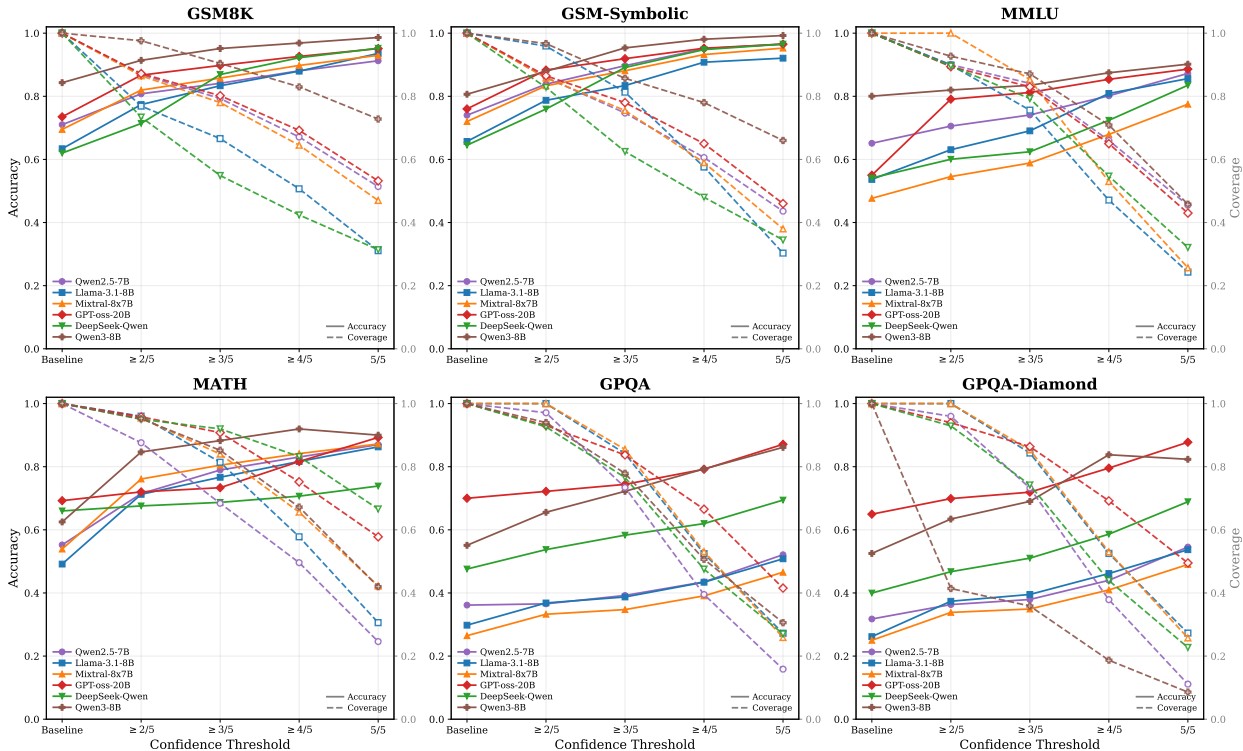

Figure 3: Coverage–accuracy trade-off across confidence thresholds. Solid lines denote accuracy and dashed lines denote coverage. As the agreement threshold increases from baseline to unanimous, accuracy improves while coverage decreases correspondingly.

Figure 3 illustrates the fundamental trade-off enabled by our method. We observe a consistent relationship between the agreement threshold and accuracy. As the threshold increases from a plurality ($\geq 2/5$ agreement) through majority requirements ($\geq 3/5$) to unanimous agreement ($\geq 5/5$), the reliability improves significantly.

Across the six language models and six benchmarks, the path agreement confidence measure enabled by NOISYCOCONUT provides a trade-off between *coverage* (the proportion of questions for which the system provides an answer) and *accuracy* (the correctness of answers output when the model does not abstain). We observe a consistent relationship between agreement threshold and accuracy, with performance improving as the threshold increases from the baseline (single-path inference) through $\geq 2$, $\geq 3$, $\geq 4$ and unanimous agreement thresholds.

Our results demonstrate that NOISYCOCONUT offers a mechanism for enhancing predictive accuracy through selective abstention on low-confidence predictions. The magnitude of improvement on mathematical benchmarks is particularly noteworthy. On GSM8K, no baseline model exceeds 75% accuracy, yet at the unanimous (5/5) agreement threshold, all models exceed 90% accuracy, with coverage ranging from 31.1% (Llama-3.1-8B) to 53.2% (gpt-oss-20B). The largest accuracy improvement is observed for DeepSeek-Qwen, which improves from 62.0% accuracy at the baseline to 95.2% at unanimous agreement, while maintaining 31.4% coverage. Similar patterns emerge across all evaluated models, indicating that path agreement constitutes an effective method for measuring prediction reliability.

**Accuracy Improvement Analysis.** Figure 4 shows the accuracy gains achieved at each confidence threshold relative to baseline performance. The heatmap reveals consistent improvements across all model–benchmark combinations, with gains ranging from approximately 10 percentage points at the lowest agreement threshold to over 40 percentage points with unanimous agreement for certain models. Models with lower baseline accuracy exhibit the largest absolute improvements, achieving near-parity with stronger mod-

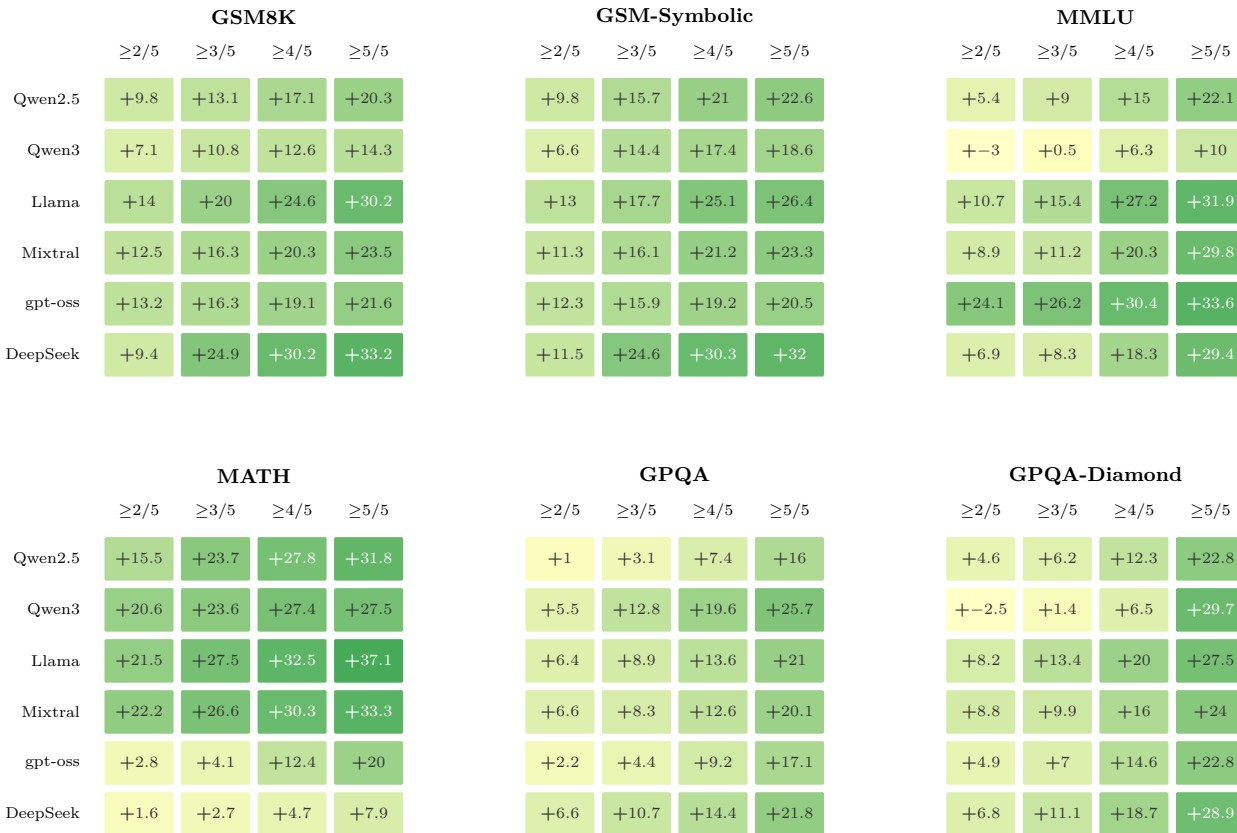

Figure 4: Cumulative accuracy improvement over baseline (in percentage points) at each confidence threshold across six benchmarks. Higher agreement requirements yield larger improvements across all models and benchmarks. Values represent the accuracy gain when the model provides an answer at that confidence threshold.

els when predictions are restricted to high-agreement instances. This finding suggests that NOISYCOCONUT can serve as an equalizing mechanism, enabling weaker models to approach the reliability of stronger counterparts on the subset of questions where they exhibit high confidence. The monotonic increase in improvement magnitude as the agreement threshold rises from $\geq 2/5$ to unanimous (5/5) agreement, visible across every column of Figure 4, further corroborates the effectiveness of agreement level as a confidence signal.

**Error Distribution.** Figure 5 presents another perspective through examination of error rates stratified by agreement level. Test examples where there is low path agreement exhibit substantially elevated error rates, typically ranging from 40% to 70%, whereas high-agreement predictions demonstrate markedly lower error rates, generally below 15% for instances of unanimous agreement. These results show consistent reduction in error rates as the agreement threshold is increasing across all benchmarks for all of the tested models.

This concentration of errors within low-agreement categories provides direct empirical validation for the assumption underlying NOISYCOCONUT: path disagreement serves as a reliable indicator of prediction uncertainty, effectively identifying instances where model outputs are unreliable and where abstention or additional verification would be warranted. The consistency of this pattern across diverse models and benchmarks underscores the robustness of agreement-based confidence estimation as a general-purpose mechanism for uncertainty quantification in language model inference.

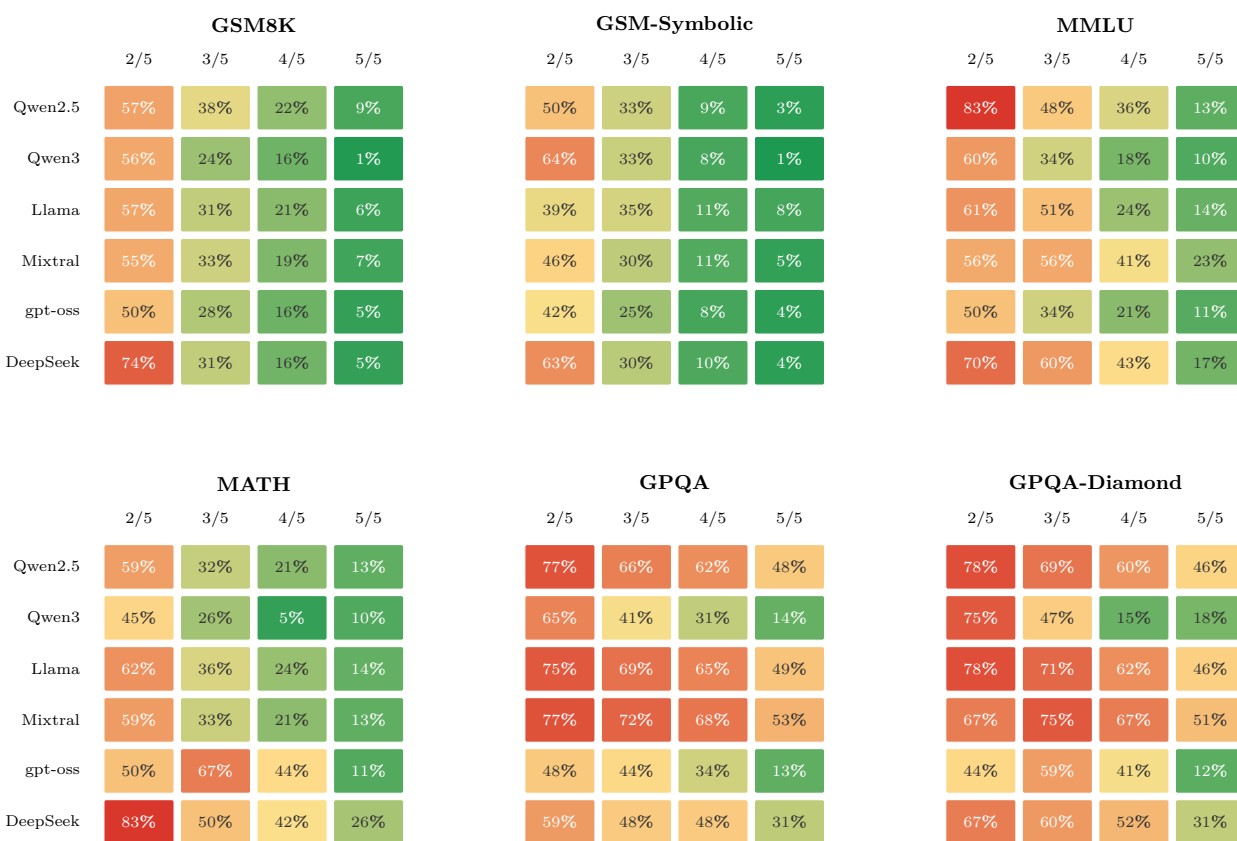

Figure 5: Error rates at each agreement level across models and benchmarks. Lower agreement levels (2/5, 3/5) consistently exhibit higher error rates (shown in red/orange), while high-agreement predictions (4/5, 5/5) achieve substantially lower error rates (shown in green). This confirms that path disagreement effectively identifies unreliable predictions suitable for abstention.

## 5 Discussion

By injecting controlled noise into hidden states, NOISYCOCONUT provides an inference-time method that introduces an accuracy–coverage tradeoff which can be used to enhance LLM reliability without any need for retraining. The consistent relationship between path agreement and accuracy suggests that uncertainty in LLMs may be visible in latent space. When noise perturbations lead to divergent reasoning paths, this indicates the model is operating in a region of representational instability where small changes in hidden states lead to substantially different outputs. Conversely, when perturbed paths converge to the same answer, the model appears to be in a stable attractor basin where the solution is robust to perturbation. This interpretation aligns with recent theoretical work analyzing how reasoning emerges through superposition of computational states in continuous representations (Zhu et al., 2025a).

The sigmoid degradation pattern observed in Figure 2, with fits achieving $R^2 \geq 0.94$ across all models, evidences a phase transition in model behavior. At low noise levels, the model's reasoning process is robust enough to absorb perturbations without changing outputs. Beyond a critical threshold, perturbations overwhelm the signal, causing rapid degradation to near-random performance. The consistency of this pattern across architectures, from instruction-tuned models to distilled reasoning models, indicates this may be a fundamental property of how transformer representations encode reasoning processes rather than an artifact of specific training procedures.

Our finding that weaker baseline models exhibit larger absolute improvements under high-agreement filtering merits further investigation. One hypothesis is that weaker models possess latent capabilities that are

inconsistently activated during standard inference. NoisyCoconut may function as a capability elicitation mechanism, identifying instances where the model "knows" the answer but requires favorable initialization to reliably surface it. This speculation connects to work on latent knowledge in language models (Christiano et al., 2021) and suggests that agreement-based filtering could complement techniques designed to extract reliable knowledge from uncertain models.

## 5.1 Limitations and Future Directions

Our experimental results provide support for the promise of the NoisyCoconut approach, but have only explored a limited space. Here, we discuss limitations of our experiments to date, and potential directions for future work.

**Cost.** A straightforward implementation of NoisyCoconut requires computational overhead to generating $K$ paths that scales approximately linearly with $K$. While we believe $K = 5$ is already a reasonable cost for improved reliability in many settings, this may be prohibitively expensive for many applications where inference costs already dominate.

Our current NoisyCoconut implementation generates paths independently, missing opportunities for computation sharing. Speculative decoding techniques (Leviathan et al., 2023; Chen et al., 2023a) demonstrate that parallel verification can be substantially cheaper than parallel generation. Analogously, architectures that share early computation across paths while branching only at key decision points could reduce overhead while maintaining diversity. Tree-structured approaches (Yao et al., 2023) provide one template, though adapting these to continuous latent space rather than discrete token space requires further development.

Techniques for early termination when paths show rapid convergence could mitigate this cost. It may also be the case that multiple executions of a smaller model that can be run locally within the NoisyCoconut framework can be used to provide more reliable results for lower cost than would be required for a single execution of an expensive proprietary model.

Controlling the computational budget allocated to reasoning is an active research area. Length Controlled Policy Optimization (LCPO) enables precise control over reasoning length, allowing models to generate outputs adhering to user-specified length constraints (Aggarwal & Madaan, 2025). Similarly, the simple test-time scaling (s1) approach introduces "budget forcing" as a technique to control test-time computation by either terminating the model's thinking process early or extending it to encourage further reasoning (Muennighoff et al., 2025). Both approaches reveal that models can adapt their reasoning strategies based on available computational resources. Combining these insights with NoisyCoconut suggests an adaptive approach: allocate more reasoning paths to instances showing early signs of disagreement while terminating early when paths rapidly converge. Such selective computation could maintain reliability gains while substantially reducing average-case overhead.

**Generalizability.** NoisyCoconut requires the ability to access and perturb internal model states, limiting applicability to open-weight models or settings with sufficient API access. The method cannot currently be applied to closed models accessible only through text-based APIs. As latent-space methods gain prominence, API designs that expose intermediate representations could enable broader application of techniques like ours (Gao et al., 2023).

Our method's effectiveness varies across architectures. Most notably, gpt-oss-20B required using first-layer hidden states rather than final-layer states for the continuous feedback loop, as the default configuration produced degenerate outputs. This sensitivity suggests that the structure of hidden representations may differ across model families, and optimal noise injection strategies may need architecture-specific tuning. Understanding why certain architectures respond differently to latent perturbation remains an open question.

**Discrete Responses.** Our evaluation focused on mathematical reasoning and knowledge-intensive tasks where the set of responses is small (multiple-choice questions) and agreement is well-defined. Extending NoisyCoconut to open-ended generation tasks where "agreement" requires semantic similarity rather than exact matching poses a difficult, but we think not impenetrable, challenge. Integration with semantic

entropy methods (Kuhn et al., 2023; Farquhar et al., 2024) could enable agreement-based confidence for tasks like summarization or translation, where multiple valid outputs exist. Additionally, exploring whether latent perturbation can improve factuality in long-form generation, where hallucination risks compound, represents a practically important direction.

**Design space exploration.** While we found $\sigma_0 = 0.2$ to be effective across our experiments, the optimal noise scale likely depends on task difficulty, model capacity, and input characteristics. Our current approach uses fixed hyperparameters, but adaptive noise scaling based on input uncertainty or model confidence could improve robustness. The exponentially weighted moving average adaptation in Equation 5 represents a first step, but more sophisticated approaches drawing on adaptive gradient methods (Kingma & Ba, 2015) or learned noise schedules (Ho et al., 2020) may prove beneficial.

Our evaluation was limited to consensus by voting as the aggregation strategy, where outputs are compared via exact match. The space of possible aggregation strategies is vast, and more sophisticated approaches such as learned aggregation functions that consider output semantics, confidence-weighted schemes based on token-level probabilities, or methods that leverage partial agreement structure may yield improved performance. Additionally, our exact-match criterion for agreement is well-suited to mathematical reasoning tasks with unique correct answers but may be overly restrictive for domains where semantically equivalent but lexically distinct outputs are valid. Integrating semantic similarity measures (Kuhn et al., 2023) or embedding-based comparison into the aggregation step could extend NoisyCoconut to open-ended generation tasks where multiple valid phrasings exist. Making the aggregation adaptive would also offer additional opportunities, using additional executions only when necessary to reduce cost, and providing finer grained methods for determining when to abstain, when to invest more compute, and when there is sufficient confidence to produce a reliable output.

**Integration with Learned Latent Reasoning.** One promising direction involves architectures specifically designed to leverage the latent space directly for extended reasoning. Recent work has introduced recurrent latent reasoning architectures that enable scaling test-time computation through iterative processing in latent space (Tan et al., 2025; Orlicki, 2025; Xu et al., 2025a). These approaches employ core recurrent blocks that can be executed multiple times before producing a final output, allowing models to perform more computation without generating additional tokens. Since these architectures explicitly optimize representations for iterative reasoning, they may exhibit even more structured uncertainty geometry that NoisyCoconut could exploit. Investigating whether our agreement signal remains predictive—or becomes even more informative—when applied to such architectures represents a natural extension.

**Language Mixing and Code-Switching Phenomena.** During our experiments with varying noise thresholds, we observed an unexpected phenomenon: at certain perturbation levels, models exhibited spontaneous language mixing, producing reasoning traces that alternated between English and other languages (e.g., Chinese) despite receiving English-only prompts (Appendix D). Intriguingly, these code-switched outputs sometimes yielded correct answers where monolingual reasoning failed, suggesting that language mixing may activate alternative reasoning pathways or access knowledge encoded differently across linguistic subspaces. This observation aligns with recent systematic studies by Wang et al. (2025), who demonstrate that language mixing in reasoning language models reflects latent processing preferences and that forcing models to reason in specific scripts can notably improve accuracy. The broader phenomenon connects to emerging research on latent chain-of-thought reasoning, which decouples reasoning from explicit language generation (Chen et al., 2025), and work showing that activation-space perturbations can encode complex reasoning patterns (Zhang & Viteri, 2025). However, we did not pursue this direction further, as we lacked a principled mechanism to predict when code-switching would be beneficial or to reliably induce it. Developing methods to detect instances where language mixing improves reasoning—and techniques to controllably trigger such behavior—represents a promising avenue for future work, potentially combining our agreement-based confidence estimation with script-aware decoding strategies.

**Theoretical Foundations.** The empirical success of agreement-based confidence estimation invites theoretical investigation. Why should path diversity in latent space correlate with correctness? Recent work by Zhu et al. (2025a) provides one lens through their analysis of reasoning via superposition, suggesting that

correct solutions may correspond to more stable superposition states. Gozeten et al. (2025) demonstrate that continuous representations enable parallel exploration of solution paths, which may explain why noise injection effectively diversifies reasoning. Developing a formal framework connecting latent geometry, perturbation stability, and output correctness could guide principled improvements to noise injection strategies and provide guarantees on when agreement-based confidence is reliable.

**Tool-augmented and agentic workflows.** A natural extension of NoisyCoconut is to use its agreement signal not only to decide whether to answer, but to decide how to act. In tool-augmented and agentic settings, a model can invoke external resources such as a calculator, a code interpreter, a retrieval system, or a search engine. Because such calls carry a cost, they are best reserved for inputs on which the model is genuinely uncertain. The disagreement among noise-perturbed latent paths provides exactly such a trigger: when the $K$ paths converge, the model can answer directly, whereas when they diverge, rather than abstaining outright (Section 3), the system can route the query to a tool, request additional context, or escalate to a stronger model or a human verifier. This reframes abstention as one option within a broader action space, turning the binary answer-or-abstain decision into a graded policy over actions. Since the confidence signal is computed entirely at inference time and requires no parameter updates, it can be layered onto existing tool-use pipelines without retraining. We leave a systematic study of such confidence-gated tool invocation, including how the agreement threshold trades the tool-call budget against accuracy, to future work.

### Acknowledgments

We thank David Onks for verifying the reproducibility of the method and preliminary work on logit-weighted confidence analysis. We are grateful to Hannah Cyberey for helpful discussions and feedback. This work is supported in part by funds provided by the National Science Foundation, Department of Homeland Security, and IBM through the ACTION AI Institute (Award #2229876) and benefited from computational resources provided by the University of Virginia Research Computing.

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

## A Path Diversity Scaling

This appendix supports the claim in Section 3.2 that the expected pairwise trajectory diversity grows with the injected noise as $\mathbb{E}[\mathcal{D}_K] = \Omega(\sigma_0\sqrt{dT})$. We first report the empirical measurements that motivated the claim (Table 2), then give a first-order derivation.

### A.1 Empirical Confirmation

Table 2 reports the mean pairwise diversity $\mathcal{D}_K$ (Equation 6) across noise scales for each model. Diversity is 0 when no noise is injected ($\sigma_0 = 0$) and increases monotonically with $\sigma_0$, as expected. The absolute magnitude of $\mathcal{D}_K$ differs substantially across models, but this is almost entirely an artifact of differing hidden-state scales: the final column normalizes $\mathcal{D}_K$ by the hidden-state magnitude at $\sigma_0 = 0.2$, and the resulting dimensionless values fall in a comparable band (0.308–0.688). In short, these measurements confirm what the analysis below predicts—$\sigma_0$ provides direct, monotone control over ensemble spread—and the residual cross-model variation is consistent with the model-dependent expansion constant introduced in the derivation.

| Model | Mean $\mathcal{D}_K$ at noise scale $\sigma_0$ | | | | | Norm. $\mathcal{D}_K$ |
| | 0.00 | 0.05 | 0.10 | 0.20 | 0.50 | ($\sigma_0$=0.2) |
|---|---|---|---|---|---|---|
| Qwen2.5-7B | 0.0 | 103.98 | 136.92 | 159.99 | 179.95 | 0.622 |
| Qwen3-8B | 0.0 | 59.1 | 77.8 | 90.9 | 102.2 | 0.688 |
| gpt-oss-20B | 0.0 | 302.4 | 398.2 | 465.3 | 523.4 | 0.593 |
| Llama-3.1-8B | 0.0 | 37.8 | 49.7 | 58.1 | 65.3 | 0.424 |
| DeepSeek-R1-7B | 0.0 | 32.2 | 42.4 | 49.5 | 55.7 | 0.308 |

Table 2: Path diversity $\mathcal{D}_K$ (Equation 6) grows with the injected noise scale $\sigma_0$, as predicted by $\mathbb{E}[\mathcal{D}_K] = \Omega(\sigma_0\sqrt{dT})$ (Section 3.2). Absolute $\mathcal{D}_K$ differs across models chiefly because of hidden-state scale. The final column reports the dimensionless normalized $\mathcal{D}_K$ at $\sigma_0 = 0.2$, which falls in a comparable band.

### A.2 Derivation

We formalize the scaling claim of Section 3.2 as Theorem 1 below. The argument is first-order: it treats the linearized dynamics (Equation 8) as the governing model, so the theorem is best read as a scaling law for the linearized regime, which is why we verify it empirically in Section A.1. It yields a *lower* bound, certifying that diversity cannot fall below this order.

**Assumption 1** (Dynamics). *The perturbed paths evolve as* $\mathbf{h}_{t+1}^{(i)} = f_\theta(\mathbf{h}_t^{(i)} + \boldsymbol{\eta}_t^{(i)})$ *with* $\boldsymbol{\eta}_t^{(i)} \sim \mathcal{N}(\mathbf{0}, \sigma_t^2\mathbf{I}_d)$ *drawn independently across paths i and steps t, and satisfy:*

(A1) *(Common initialization)* $\mathbf{h}_0^{(i)} = \mathbf{h}_0$ *for all i.*

(A2) *(Local expansiveness) There is a constant* $c \geq 1$ *such that the Jacobian* $\mathbf{J}_t = \nabla f_\theta(\mathbf{m}_t)$, *evaluated at the midpoint* $\mathbf{m}_t = \frac{1}{2}(\mathbf{h}_t^{(i)} + \mathbf{h}_t^{(j)}) + \frac{1}{2}(\boldsymbol{\eta}_t^{(i)} + \boldsymbol{\eta}_t^{(j)})$, *obeys* $\mathbb{E}\|\mathbf{J}_t\mathbf{v}\|_2^2 \geq c^2\|\mathbf{v}\|_2^2$ *for every* $\mathbf{v} \in \{\boldsymbol{\delta}_t, \boldsymbol{\xi}_t, \boldsymbol{\delta}_t + \boldsymbol{\xi}_t\}$, *with* $\boldsymbol{\delta}_t, \boldsymbol{\xi}_t$ *as in the proof.*

(A3) *(Noise floor)* $\sigma_t \geq \sigma_{\min} > 0$ *over the steps considered, which holds for moderate* $\lambda T$ *since* $\sigma_t = \sigma_0 e^{-\lambda t}\|\mathbf{h}_t\|_2/\mu_t$.

(A4) *(Anti-concentration) There is a constant* $\kappa \in (0, 1]$ *with* $\mathbb{E}\|\boldsymbol{\delta}_t\|_2 \geq \kappa\sqrt{\mathbb{E}\|\boldsymbol{\delta}_t\|_2^2}$ *for all t. This holds whenever* $\boldsymbol{\delta}_t$ *has bounded kurtosis; for jointly Gaussian* $\boldsymbol{\delta}_t$ *one may take* $\kappa \to 1$ *as* $d \to \infty$.

**Theorem 1.** *Under Assumption 1 and the first-order linearization in Equation 8, the expected pairwise trajectory diversity (Equation 6) satisfies*

$$\mathbb{E}[\mathcal{D}_K] = \Omega\left(\sigma_0\sqrt{dT}\right).$$

*Proof.* Fix two distinct paths $i \neq j$ and set $\boldsymbol{\delta}_t = \mathbf{h}_t^{(i)} - \mathbf{h}_t^{(j)}$, so that $\boldsymbol{\delta}_0 = \mathbf{0}$ by (A1). We bound $\mathbb{E}\|\boldsymbol{\delta}_t\|_2$ in four steps and then average.

*Step 1 (Single-step recursion).* At step $t$ the inputs to $f_\theta$ for the two paths differ by $\boldsymbol{\delta}_t + \boldsymbol{\xi}_t$, where $\boldsymbol{\xi}_t := \boldsymbol{\eta}_t^{(i)} - \boldsymbol{\eta}_t^{(j)} \sim \mathcal{N}(\mathbf{0}, 2\sigma_t^2 \mathbf{I}_d)$ is independent of $\boldsymbol{\delta}_t$. Linearizing $f_\theta$ about $\mathbf{m}_t$,

$$\boldsymbol{\delta}_{t+1} = \mathbf{J}_t(\boldsymbol{\delta}_t + \boldsymbol{\xi}_t). \tag{8}$$

Taking expectations of the squared norm and using $\mathbb{E}[\boldsymbol{\xi}_t] = \mathbf{0}$ together with the independence of $\boldsymbol{\xi}_t$ and $\boldsymbol{\delta}_t$, the cross term vanishes, so

$$\mathbb{E}\|\boldsymbol{\delta}_{t+1}\|_2^2 = \mathbb{E}\|\mathbf{J}_t\boldsymbol{\delta}_t\|_2^2 + \mathbb{E}\|\mathbf{J}_t\boldsymbol{\xi}_t\|_2^2 \geq c^2\,\mathbb{E}\|\boldsymbol{\delta}_t\|_2^2 + 2c^2\sigma_t^2 d, \tag{9}$$

where the inequality applies (A2) to both terms and uses $\mathbb{E}\|\boldsymbol{\xi}_t\|_2^2 = 2\sigma_t^2 d$.

*Step 2 (Accumulation).* Unrolling equation 9 from $\boldsymbol{\delta}_0 = \mathbf{0}$,

$$\mathbb{E}\|\boldsymbol{\delta}_t\|_2^2 \geq 2c^2 d \sum_{s=0}^{t-1} c^{2(t-1-s)}\sigma_s^2. \tag{10}$$

Since $c \geq 1$, each factor $c^{2(t-1-s)} \geq 1$; with the noise floor $\sigma_s \geq \sigma_{\min}$ from (A3),

$$\mathbb{E}\|\boldsymbol{\delta}_t\|_2^2 \geq 2c^2 d \sum_{s=0}^{t-1} \sigma_s^2 \geq 2c^2 d\, t\, \sigma_{\min}^2. \tag{11}$$

*Step 3 (Second moment to first moment).* Applying (A4) to equation 11,

$$\mathbb{E}\|\boldsymbol{\delta}_t\|_2 \geq \kappa\sqrt{\mathbb{E}\|\boldsymbol{\delta}_t\|_2^2} \geq \kappa\, c\, \sigma_{\min}\sqrt{2d}\,\sqrt{t}. \tag{12}$$

*Step 4 (Averaging).* The diversity equation 6 averages $\|\boldsymbol{\delta}_t\|_2$ over the $T$ steps and over the $\binom{K}{2}$ identically distributed pairs. Averaging equation 12 over $t$ and using $\frac{1}{T}\sum_{t=0}^{T-1}\sqrt{t} \geq \frac{2}{3}\sqrt{T}\,(1 - o(1))$,

$$\mathbb{E}[\mathcal{D}_K] = \frac{1}{T}\sum_{t=0}^{T-1}\mathbb{E}\|\boldsymbol{\delta}_t\|_2 \geq \frac{2\kappa c}{3}\sigma_{\min}\sqrt{2dT}\,(1 - o(1)) = \Omega\left(\sigma_{\min}\sqrt{dT}\right). \tag{13}$$

Finally, $\sigma_{\min}/\sigma_0 = e^{-\lambda(T-1)}\|\mathbf{h}_{T-1}\|_2/\mu_{T-1}$ is bounded below by a positive constant for moderate $\lambda T$, whence $\Omega(\sigma_{\min}\sqrt{dT}) = \Omega(\sigma_0\sqrt{dT})$. $\qquad\square$

**Discussion.** The result depends critically on the local-expansiveness hypothesis $c \geq 1$ in (A2). In the strongly contractive regime $c < 1$, the propagation factors $c^{2(t-1-s)}$ in equation 10 attenuate the accumulated noise and the bound degrades to $\Omega(\sigma_{\min}\sqrt{d})$ with no growth in $T$. This is consistent with the cross-model variation in the normalized $\mathcal{D}_K$ reported in Table 2 (0.308–0.688), which reflects differing effective expansion constants $c$ across architectures.

# B   Detailed Results

This appendix reports the full per-benchmark breakdown of NOISYCOCONUT across all six benchmarks and six models, stratified by agreement outcome. Tables 3–8 give the question counts, correct/incorrect/abstention tallies, and accuracy at each agreement tier.

|  |  | Question Count | Correct | No Answer | Incorrect | Accuracy (%) |
|---|---|---|---|---|---|---|
| Qwen2.5-7B | Baseline (No majority) | 1000 | 639 | 0 | 261 | 71.00 |
|  | Minimal Majority (2/5) | 54 | 23 | 0 | 31 | 42.59 |
|  | Moderate Majority (3/5) | 122 | 76 | 0 | 46 | 62.30 |
|  | Other | 31 | 2 | 0 | 29 | 6.45 |
|  | Split Vote (2/2) | 22 | 12 | 0 | 10 | 54.55 |
|  | Strong Majority (4/5) | 157 | 122 | 0 | 35 | 77.71 |
|  | Unanimous (5/5) | 514 | 469 | 0 | 45 | 91.25 |
| Llama-3.1-8B | Baseline (No majority) | 1000 | 507 | 0 | 293 | 63.38 |
|  | Minimal Majority (2/5) | 69 | 30 | 0 | 39 | 43.48 |
|  | Moderate Majority (3/5) | 159 | 109 | 0 | 50 | 68.55 |
|  | Other | 31 | 1 | 0 | 30 | 3.23 |
|  | Split Vote (2/2) | 34 | 10 | 0 | 24 | 29.41 |
|  | Strong Majority (4/5) | 196 | 155 | 0 | 41 | 79.08 |
|  | Unanimous (5/5) | 311 | 291 | 0 | 20 | 93.57 |
| Mixtral-8x7B | Baseline (No majority) | 1000 | 695 | 0 | 305 | 69.50 |
|  | Minimal Majority (2/5) | 58 | 26 | 0 | 32 | 44.83 |
|  | Moderate Majority (3/5) | 135 | 90 | 0 | 45 | 66.67 |
|  | Other | 35 | 3 | 0 | 32 | 8.57 |
|  | Split Vote (2/2) | 27 | 14 | 0 | 13 | 51.85 |
|  | Strong Majority (4/5) | 175 | 142 | 0 | 33 | 81.14 |
|  | Unanimous (5/5) | 470 | 437 | 0 | 33 | 92.98 |
| gpt-oss-20B | Baseline (No majority) | 1000 | 735 | 0 | 265 | 73.50 |
|  | Minimal Majority (2/5) | 50 | 25 | 0 | 25 | 50.00 |
|  | Moderate Majority (3/5) | 110 | 79 | 0 | 31 | 71.82 |
|  | Other | 28 | 4 | 0 | 24 | 14.29 |
|  | Split Vote (2/2) | 20 | 11 | 0 | 9 | 55.00 |
|  | Strong Majority (4/5) | 160 | 135 | 0 | 25 | 84.38 |
|  | Unanimous (5/5) | 532 | 506 | 0 | 26 | 95.11 |
| DeepSeek-Qwen | Baseline (No majority) | 1000 | 496 | 0 | 304 | 62.00 |
|  | Minimal Majority (2/5) | 142 | 37 | 0 | 105 | 26.06 |
|  | Moderate Majority (3/5) | 125 | 86 | 0 | 39 | 68.80 |
|  | Other | 66 | 13 | 0 | 53 | 19.70 |
|  | Split Vote (2/2) | 43 | 10 | 0 | 33 | 23.26 |
|  | Strong Majority (4/5) | 110 | 92 | 0 | 18 | 83.64 |
|  | Unanimous (5/5) | 314 | 299 | 0 | 15 | 95.22 |
| Qwen3-8B | Baseline (No majority) | 1000 | 838 | 6 | 156 | 84.31 |
|  | Minimal Majority (2/5) | 50 | 22 | 0 | 28 | 44.00 |
|  | Moderate Majority (3/5) | 76 | 56 | 2 | 18 | 75.68 |
|  | Other | 22 | 4 | 0 | 18 | 18.18 |
|  | Split Vote (2/2) | 22 | 10 | 0 | 12 | 45.45 |
|  | Strong Majority (4/5) | 102 | 86 | 0 | 16 | 84.31 |
|  | Unanimous (5/5) | 728 | 718 | 0 | 10 | 98.63 |

Table 3: GSM8K Results.

| | | Question Count | Correct | No Answer | Incorrect | Accuracy (%) |
|---|---|---|---|---|---|---|
| Qwen2.5-7B | Baseline (No majority) | 1000 | 666 | 0 | 234 | 74.00 |
| | Minimal Majority (2/5) | 82 | 41 | 0 | 41 | 50.00 |
| | Moderate Majority (3/5) | 141 | 94 | 0 | 47 | 66.67 |
| | Other | 40 | 3 | 0 | 37 | 7.50 |
| | Split Vote (2/2) | 31 | 10 | 0 | 21 | 32.26 |
| | Strong Majority (4/5) | 170 | 155 | 0 | 15 | 91.18 |
| | Unanimous (5/5) | 436 | 421 | 0 | 15 | 96.56 |
| Llama-3.1-8B | Baseline (No majority) | 1000 | 657 | 0 | 343 | 65.70 |
| | Minimal Majority (2/5) | 104 | 64 | 0 | 40 | 61.54 |
| | Moderate Majority (3/5) | 237 | 155 | 0 | 82 | 65.40 |
| | Other | 41 | 10 | 0 | 31 | 24.39 |
| | Split Vote (2/2) | 42 | 13 | 0 | 29 | 30.95 |
| | Strong Majority (4/5) | 273 | 244 | 0 | 29 | 89.38 |
| | Unanimous (5/5) | 303 | 279 | 0 | 24 | 92.08 |
| Mixtral-8x7B | Baseline (No majority) | 1000 | 720 | 0 | 280 | 72.00 |
| | Minimal Majority (2/5) | 70 | 38 | 0 | 32 | 54.29 |
| | Moderate Majority (3/5) | 165 | 115 | 0 | 50 | 69.70 |
| | Other | 42 | 5 | 0 | 37 | 11.90 |
| | Split Vote (2/2) | 33 | 12 | 0 | 21 | 36.36 |
| | Strong Majority (4/5) | 210 | 188 | 0 | 22 | 89.52 |
| | Unanimous (5/5) | 380 | 362 | 0 | 18 | 95.26 |
| gpt-oss-20B | Baseline (No majority) | 1000 | 760 | 0 | 240 | 76.00 |
| | Minimal Majority (2/5) | 60 | 35 | 0 | 25 | 58.33 |
| | Moderate Majority (3/5) | 130 | 98 | 0 | 32 | 75.38 |
| | Other | 35 | 7 | 0 | 28 | 20.00 |
| | Split Vote (2/2) | 25 | 12 | 0 | 13 | 48.00 |
| | Strong Majority (4/5) | 190 | 175 | 0 | 15 | 92.11 |
| | Unanimous (5/5) | 460 | 444 | 0 | 16 | 96.52 |
| DeepSeek-Qwen | Baseline (No majority) | 1000 | 645 | 0 | 355 | 64.50 |
| | Minimal Majority (2/5) | 155 | 58 | 0 | 97 | 37.42 |
| | Moderate Majority (3/5) | 145 | 102 | 0 | 43 | 70.34 |
| | Other | 72 | 18 | 0 | 54 | 25.00 |
| | Split Vote (2/2) | 48 | 14 | 0 | 34 | 29.17 |
| | Strong Majority (4/5) | 135 | 122 | 0 | 13 | 90.37 |
| | Unanimous (5/5) | 345 | 333 | 0 | 12 | 96.52 |
| Qwen3-8B | Baseline (No majority) | 1000 | 802 | 5 | 193 | 80.65 |
| | Minimal Majority (2/5) | 73 | 25 | 3 | 45 | 35.71 |
| | Moderate Majority (3/5) | 80 | 52 | 3 | 25 | 67.74 |
| | Other | 27 | 5 | 0 | 22 | 18.18 |
| | Split Vote (2/2) | 40 | 8 | 0 | 32 | 18.75 |
| | Strong Majority (4/5) | 120 | 110 | 0 | 10 | 91.67 |
| | Unanimous (5/5) | 660 | 655 | 0 | 5 | 99.24 |

Table 4: GSM-Symbolic Results.

|  |  | Question Count | Correct | No Answer | Incorrect | Accuracy (%) |
|---|---|---|---|---|---|---|
| Qwen2.5-7B | Baseline (No majority) | 1000 | 586 | 0 | 314 | 65.11 |
|  | Minimal Majority (2/5) | 12 | 2 | 0 | 10 | 16.67 |
|  | Moderate Majority (3/5) | 181 | 94 | 0 | 87 | 51.93 |
|  | Split Vote (2/2) | 47 | 10 | 0 | 37 | 21.28 |
|  | Strong Majority (4/5) | 205 | 132 | 0 | 73 | 64.39 |
|  | Unanimous (5/5) | 455 | 397 | 0 | 58 | 87.25 |
| Llama-3.1-8B | Baseline (No majority) | 1000 | 483 | 0 | 417 | 53.67 |
|  | Minimal Majority (2/5) | 41 | 16 | 0 | 25 | 39.02 |
|  | Moderate Majority (3/5) | 285 | 141 | 0 | 144 | 49.47 |
|  | Other | 1 | 0 | 0 | 1 | 0.00 |
|  | Split Vote (2/2) | 102 | 29 | 0 | 73 | 28.43 |
|  | Strong Majority (4/5) | 228 | 173 | 0 | 55 | 75.88 |
|  | Unanimous (5/5) | 243 | 208 | 0 | 35 | 85.60 |
| Mixtral-8x7B | Baseline (No majority) | 900 | 429 | 0 | 471 | 47.67 |
|  | Minimal Majority (2/5) | 27 | 12 | 0 | 15 | 44.44 |
|  | Moderate Majority (3/5) | 291 | 128 | 0 | 163 | 43.99 |
|  | Split Vote (2/2) | 105 | 27 | 0 | 78 | 25.71 |
|  | Strong Majority (4/5) | 246 | 145 | 0 | 101 | 58.94 |
|  | Unanimous (5/5) | 231 | 179 | 0 | 52 | 77.49 |
| gpt-oss-20B | Baseline (No majority) | 1000 | 550 | 0 | 450 | 55.00 |
|  | Minimal Majority (2/5) | 12 | 6 | 0 | 6 | 50.00 |
|  | Moderate Majority (3/5) | 180 | 119 | 0 | 61 | 66.11 |
|  | Other | 6 | 1 | 0 | 5 | 16.67 |
|  | Split Vote (2/2) | 52 | 27 | 0 | 25 | 51.92 |
|  | Strong Majority (4/5) | 220 | 174 | 0 | 46 | 79.09 |
|  | Unanimous (5/5) | 430 | 381 | 0 | 49 | 88.60 |
| DeepSeek-Qwen | Baseline (No majority) | 1000 | 487 | 0 | 413 | 54.11 |
|  | Minimal Majority (2/5) | 27 | 8 | 0 | 19 | 29.63 |
|  | Moderate Majority (3/5) | 246 | 99 | 0 | 147 | 40.24 |
|  | Other | 4 | 0 | 0 | 4 | 0.00 |
|  | Split Vote (2/2) | 76 | 35 | 0 | 41 | 46.05 |
|  | Strong Majority (4/5) | 226 | 128 | 0 | 98 | 56.64 |
|  | Unanimous (5/5) | 321 | 268 | 0 | 53 | 83.49 |
| Qwen3-8B | Baseline (No majority) | 1000 | 708 | 116 | 176 | 80.04 |
|  | Minimal Majority (2/5) | 5 | 2 | 0 | 3 | 33.33 |
|  | Moderate Majority (3/5) | 200 | 107 | 38 | 55 | 66.29 |
|  | Other | 0 | 0 | 0 | 0 | – |
|  | Split Vote (2/2) | 55 | 31 | 4 | 20 | 60.71 |
|  | Strong Majority (4/5) | 273 | 207 | 22 | 44 | 82.61 |
|  | Unanimous (5/5) | 467 | 413 | 9 | 45 | 90.08 |

Table 5: MMLU Results

| | | Question Count | Correct | No Answer | Incorrect | Accuracy (%) |
|---|---|---|---|---|---|---|
| **Qwen2.5-7B** | Baseline (No majority) | 500 | 248 | 51 | 201 | 55.23 |
| | Minimal Majority (2/5) | 71 | 27 | 6 | 38 | 41.54 |
| | Moderate Majority (3/5) | 101 | 64 | 7 | 30 | 68.09 |
| | Other | 43 | 0 | 4 | 39 | 0.00 |
| | Split Vote (2/2) | 34 | 16 | 3 | 15 | 51.61 |
| | Strong Majority (4/5) | 127 | 99 | 2 | 26 | 79.20 |
| | Unanimous (5/5) | 124 | 107 | 1 | 16 | 86.99 |
| **Llama-3.1-8B** | Baseline (No majority) | 500 | 246 | 0 | 254 | 49.15 |
| | Minimal Majority (2/5) | 52 | 20 | 0 | 32 | 37.80 |
| | Moderate Majority (3/5) | 118 | 76 | 0 | 42 | 64.00 |
| | Other | 20 | 0 | 0 | 20 | 0.00 |
| | Split Vote (2/2) | 21 | 10 | 0 | 11 | 47.48 |
| | Strong Majority (4/5) | 136 | 104 | 0 | 32 | 76.43 |
| | Unanimous (5/5) | 153 | 132 | 0 | 21 | 86.12 |
| **Mixtral-8x7B** | Baseline (No majority) | 500 | 270 | 0 | 230 | 53.90 |
| | Minimal Majority (2/5) | 39 | 16 | 0 | 23 | 40.77 |
| | Moderate Majority (3/5) | 92 | 62 | 0 | 30 | 67.38 |
| | Other | 23 | 0 | 0 | 23 | 0.00 |
| | Split Vote (2/2) | 18 | 9 | 0 | 9 | 50.79 |
| | Strong Majority (4/5) | 117 | 92 | 0 | 25 | 78.90 |
| | Unanimous (5/5) | 211 | 184 | 0 | 27 | 87.25 |
| **gpt-oss-20B** | Baseline (No majority) | 500 | 337 | 14 | 149 | 69.24 |
| | Minimal Majority (2/5) | 10 | 5 | 0 | 5 | 46.10 |
| | Moderate Majority (3/5) | 82 | 26 | 4 | 52 | 32.83 |
| | Other | 6 | 0 | 0 | 6 | 0.00 |
| | Split Vote (2/2) | 16 | 7 | 1 | 8 | 45.12 |
| | Strong Majority (4/5) | 90 | 49 | 3 | 38 | 56.85 |
| | Unanimous (5/5) | 296 | 258 | 7 | 31 | 89.39 |
| **DeepSeek-Qwen** | Baseline (No majority) | 500 | 314 | 24 | 162 | 65.99 |
| | Minimal Majority (2/5) | 6 | 1 | 0 | 5 | 20.00 |
| | Moderate Majority (3/5) | 48 | 22 | 4 | 22 | 50.20 |
| | Other | 1 | 0 | 0 | 1 | 0.00 |
| | Split Vote (2/2) | 9 | 4 | 0 | 5 | 45.12 |
| | Strong Majority (4/5) | 89 | 48 | 6 | 35 | 57.48 |
| | Unanimous (5/5) | 347 | 246 | 14 | 87 | 73.72 |
| **Qwen3-8B** | Baseline (No majority) | 500 | 300 | 20 | 180 | 62.50 |
| | Minimal Majority (2/5) | 30 | 16 | 1 | 13 | 55.17 |
| | Moderate Majority (3/5) | 95 | 67 | 5 | 23 | 74.44 |
| | Other | 8 | 0 | 1 | 7 | 0.00 |
| | Split Vote (2/2) | 22 | 11 | 1 | 10 | 52.38 |
| | Strong Majority (4/5) | 130 | 120 | 4 | 6 | 95.24 |
| | Unanimous (5/5) | 215 | 189 | 5 | 21 | 90.00 |

Table 6: MATH Results.

| | | Question Count | Correct | No Answer | Incorrect | Accuracy (%) |
|---|---|---|---|---|---|---|
| **Qwen2.5-7B** | Baseline (No majority) | 448 | 154 | 22 | 272 | 36.15 |
| | Minimal Majority (2/5) | 38 | 8 | 4 | 26 | 23.53 |
| | Moderate Majority (3/5) | 156 | 52 | 4 | 100 | 34.21 |
| | Other | 1 | 0 | 1 | 0 | – |
| | Split Vote (2/2) | 76 | 22 | 4 | 50 | 30.56 |
| | Strong Majority (4/5) | 106 | 40 | 0 | 66 | 37.74 |
| | Unanimous (5/5) | 71 | 37 | 0 | 34 | 52.11 |
| **Llama-3.1-8B** | Baseline (No majority) | 448 | 133 | 0 | 315 | 29.79 |
| | Minimal Majority (2/5) | 20 | 5 | 0 | 15 | 25.00 |
| | Moderate Majority (3/5) | 142 | 44 | 0 | 98 | 30.87 |
| | Other | 0 | 0 | 0 | 0 | 0.00 |
| | Split Vote (2/2) | 51 | 14 | 0 | 37 | 26.62 |
| | Strong Majority (4/5) | 113 | 40 | 0 | 73 | 35.54 |
| | Unanimous (5/5) | 122 | 62 | 0 | 60 | 51.12 |
| **Mixtral-8x7B** | Baseline (No majority) | 448 | 119 | 0 | 329 | 26.46 |
| | Minimal Majority (2/5) | 13 | 3 | 0 | 10 | 25.00 |
| | Moderate Majority (3/5) | 145 | 40 | 0 | 105 | 27.71 |
| | Other | 0 | 0 | 0 | 0 | 0.00 |
| | Split Vote (2/2) | 52 | 13 | 0 | 39 | 25.00 |
| | Strong Majority (4/5) | 122 | 39 | 0 | 83 | 32.04 |
| | Unanimous (5/5) | 116 | 54 | 0 | 62 | 46.27 |
| **gpt-oss-20B** | Baseline (No majority) | 448 | 295 | 27 | 126 | 70.00 |
| | Minimal Majority (2/5) | 27 | 13 | 2 | 12 | 50.29 |
| | Moderate Majority (3/5) | 81 | 43 | 4 | 34 | 55.51 |
| | Other | 0 | 0 | 0 | 0 | 0.00 |
| | Split Vote (2/2) | 18 | 9 | 1 | 8 | 50.29 |
| | Strong Majority (4/5) | 125 | 74 | 13 | 38 | 66.48 |
| | Unanimous (5/5) | 197 | 162 | 11 | 24 | 87.17 |
| **DeepSeek-Qwen** | Baseline (No majority) | 448 | 199 | 30 | 219 | 47.58 |
| | Minimal Majority (2/5) | 33 | 13 | 1 | 19 | 39.87 |
| | Moderate Majority (3/5) | 139 | 68 | 9 | 62 | 52.15 |
| | Other | 3 | 0 | 0 | 3 | 0.00 |
| | Split Vote (2/2) | 42 | 10 | 2 | 30 | 24.99 |
| | Strong Majority (4/5) | 97 | 48 | 5 | 44 | 52.38 |
| | Unanimous (5/5) | 134 | 84 | 13 | 37 | 69.25 |
| **Qwen3-8B** | Baseline (No majority) | 448 | 233 | 25 | 190 | 55.10 |
| | Minimal Majority (2/5) | 35 | 12 | 1 | 22 | 35.29 |
| | Moderate Majority (3/5) | 130 | 72 | 8 | 50 | 59.02 |
| | Other | 2 | 0 | 0 | 2 | 0.00 |
| | Split Vote (2/2) | 40 | 12 | 2 | 26 | 31.58 |
| | Strong Majority (4/5) | 95 | 62 | 5 | 28 | 68.89 |
| | Unanimous (5/5) | 146 | 118 | 9 | 19 | 86.13 |

Table 7: GPQA Results.

|  |  | Question Count | Correct | No Answer | Incorrect | Accuracy (%) |
|---|---|---|---|---|---|---|
| **Qwen2.5-7B** | Baseline (No majority) | 198 | 60 | 9 | 129 | 31.75 |
| | Minimal Majority (2/5) | 12 | 2 | 3 | 7 | 22.22 |
| | Moderate Majority (3/5) | 71 | 22 | 1 | 48 | 31.43 |
| | Other | 2 | 0 | 0 | 2 | 0.00 |
| | Split Vote (2/2) | 38 | 12 | 2 | 24 | 33.33 |
| | Strong Majority (4/5) | 53 | 21 | 0 | 32 | 39.62 |
| | Unanimous (5/5) | 22 | 12 | 0 | 10 | 54.55 |
| **Llama-3.1-8B** | Baseline (No majority) | 198 | 52 | 0 | 146 | 26.16 |
| | Minimal Majority (2/5) | 9 | 2 | 0 | 7 | 25.00 |
| | Moderate Majority (3/5) | 63 | 18 | 0 | 45 | 28.37 |
| | Other | 0 | 0 | 0 | 0 | 0.00 |
| | Split Vote (2/2) | 22 | 6 | 0 | 16 | 29.03 |
| | Strong Majority (4/5) | 50 | 19 | 0 | 31 | 37.31 |
| | Unanimous (5/5) | 54 | 29 | 0 | 25 | 53.51 |
| **Mixtral-8x7B** | Baseline (No majority) | 198 | 50 | 0 | 148 | 25.00 |
| | Minimal Majority (2/5) | 6 | 2 | 0 | 4 | 25.00 |
| | Moderate Majority (3/5) | 64 | 16 | 0 | 48 | 25.46 |
| | Other | 0 | 0 | 0 | 0 | 0.00 |
| | Split Vote (2/2) | 23 | 6 | 0 | 17 | 25.96 |
| | Strong Majority (4/5) | 54 | 18 | 0 | 36 | 33.64 |
| | Unanimous (5/5) | 51 | 25 | 0 | 26 | 48.44 |
| **gpt-oss-20B** | Baseline (No majority) | 198 | 121 | 11 | 66 | 64.96 |
| | Minimal Majority (2/5) | 10 | 5 | 1 | 4 | 56.27 |
| | Moderate Majority (3/5) | 38 | 14 | 4 | 20 | 40.76 |
| | Other | 0 | 0 | 0 | 0 | 0.00 |
| | Split Vote (2/2) | 6 | 2 | 0 | 4 | 33.33 |
| | Strong Majority (4/5) | 42 | 23 | 3 | 16 | 58.95 |
| | Unanimous (5/5) | 102 | 86 | 4 | 12 | 86.74 |
| **DeepSeek-Qwen** | Baseline (No majority) | 198 | 74 | 14 | 110 | 39.97 |
| | Minimal Majority (2/5) | 12 | 4 | 0 | 8 | 37.40 |
| | Moderate Majority (3/5) | 63 | 24 | 3 | 36 | 39.69 |
| | Other | 0 | 0 | 0 | 0 | 0.00 |
| | Split Vote (2/2) | 26 | 7 | 1 | 18 | 29.91 |
| | Strong Majority (4/5) | 47 | 20 | 5 | 22 | 46.56 |
| | Unanimous (5/5) | 50 | 31 | 5 | 14 | 69.75 |
| **Qwen3-8B** | Baseline (No majority) | 198 | 52 | 99 | 47 | 52.53 |
| | Minimal Majority (2/5) | 9 | 1 | 5 | 3 | 25.00 |
| | Moderate Majority (3/5) | 58 | 18 | 24 | 16 | 52.94 |
| | Other | 1 | 0 | 0 | 1 | 0.00 |
| | Split Vote (2/2) | 12 | 2 | 5 | 5 | 28.57 |
| | Strong Majority (4/5) | 55 | 17 | 35 | 3 | 85.00 |
| | Unanimous (5/5) | 63 | 14 | 46 | 3 | 82.35 |

Table 8: GPQA-Diamond Results.

## C  Ablations

To validate our key hyperparameter choices, we conducted ablation studies on the two parameters that fundamentally define Noisy Coconut: the noise scale ($\sigma_0$) and the number of reasoning paths ($K$). Experiments were conducted on GSM8K using Qwen2.5-7B-Instruct. Table 9 summarizes our findings.

Table 9: Core ablation study on GSM8K (Qwen2.5-7B-Instruct, $N$=1000). We vary noise scale and path count independently. Baseline uses $K$=1 with no noise injection.

| Configuration | Accuracy (%) | $\Delta$ |
|---|---|---|
| Baseline ($K$=1, $\sigma_0$=0) | 71.0 | — |
| *Noise scale (fixed $K$=5):* | | |
| $\quad \sigma_0$=0.0 | 76.4 | +5.4 |
| $\quad \sigma_0$=0.2 | 78.2 | +7.2 |
| $\quad \sigma_0$=0.5 | 77.8 | +6.8 |
| *Path count (fixed $\sigma_0$=0.2):* | | |
| $\quad K$=1 | 72.0 | +1.0 |
| $\quad K$=5 ($\geq$3/5) | 78.2 | +7.2 |
| $\quad K$=10 ($\geq$6/10) | 80.2 | +9.2 |

Table 10: Self-consistency versus NoisyCoconut on 100 GSM8K questions (Qwen2.5-7B-Instruct, $\sigma = 0.2$, $K = 5$). Accuracy is computed over answered questions within each agreement tier. NoisyCoconut spreads mass into lower-confidence tiers while remaining more accurate at the top tiers.

| | Self-Consistency | | NoisyCoconut | |
|---|---|---|---|---|
| Agreement level | Count | Acc. (%) | Count | Acc. (%) |
| Baseline (single path) | 100 | 87.00 | 100 | 79.00 |
| No Consensus (1/5) | 1 | 0.00 | – | – |
| Minimal Majority (2/5) | – | – | 1 | 100.00 |
| Split Vote (2/2) | – | – | 1 | 0.00 |
| Moderate Majority (3/5) | 4 | 75.00 | 10 | 70.00 |
| Strong Majority (4/5) | 10 | 80.00 | 39 | 84.62 |
| Unanimous (5/5) | 85 | 90.59 | 49 | 93.88 |
| Overall (majority vote) | 100 | 88.00 | 100 | 87.00 |

Table 11: Noise decay schedule ablation on a preliminary set of 100 MMLU questions (Qwen2.5-7B-Instruct, $\sigma = 0.2$, $K = 5$). Adaptive exponential decay improves accuracy over a cosine schedule at every agreement level, with the largest gains at the lower-confidence tiers where the agreement signal is most sensitive to the schedule.

| Agreement level | Cosine (%) | Adaptive (%) | $\Delta$ (pp) |
|---|---|---|---|
| Baseline (single path) | 55.00 | 65.10 | +10.10 |
| Split Vote (2/2) | 0.00 | 21.30 | +21.30 |
| Moderate Majority (3/5) | 30.30 | 51.90 | +21.60 |
| Strong Majority (4/5) | 72.50 | 75.00 | +2.50 |
| Unanimous (5/5) | 84.00 | 91.25 | +7.25 |

### C.1  Noise Scale and Path Count

**Noise scale.** We evaluated $\sigma_0 \in \{0, 0.2, 0.5\}$ with $K$=5 paths. Setting $\sigma_0$=0 (i.e., multiple paths without noise injection) yields 76.4% accuracy (+5.4% over baseline), demonstrating that diversity alone provides substantial improvements through majority voting. However, performance peaks at $\sigma_0$=0.2 with 78.2% accuracy, indicating that noise injection provides an additional +1.8% gain beyond path diversity alone. Higher noise scales show logarithmic decay, with $\sigma_0$=0.5 achieving 77.8% as individual path quality degrades. We select $\sigma_0$=0.2 as it achieves the optimal balance between exploration diversity and reasoning coherence.

**Number of paths.** We evaluated $K \in \{1, 5, 10\}$ with $\sigma_0 = 0.2$. Performance scales sublinearly with $K$, showing diminishing returns beyond $K=5$. While $K=10$ achieves 2.0 points higher accuracy, it requires $2\times$ the computational cost. We select $K=5$ as a practical tradeoff between performance and efficiency.

## C.2  Agreement-Tier Analysis vs. Self-Consistency

To understand *how* Noisy Coconut differs from standard self-consistency, we bucket the 100 GSM8K questions by the level of agreement among the $K=5$ sampled paths and report per-tier accuracy in Table 10. Two effects stand out. First, NoisyCoconut redistributes probability mass away from the unanimous tier (85 vs. 49 questions) into the intermediate-agreement tiers, reflecting the additional exploration induced by noise injection. Second, despite this spread, NoisyCoconut is *more* accurate at the highest-confidence tiers, reaching 93.88% on unanimous answers (vs. 90.59%) and 84.62% on strong majorities (vs. 80.00%). In other words, the diversity does not come at the cost of reliability when the model is confident: when paths agree, that agreement is more trustworthy. The overall majority-vote accuracy remains comparable (87.00% vs. 88.00%), but the richer agreement signal is useful for downstream confidence estimation and selective prediction.

## C.3  Noise Decay Schedule

Finally, we ablate the schedule used to anneal the injected noise over the reasoning trajectory, comparing a standard cosine decay against our adaptive exponential decay on a preliminary set of 100 MMLU questions (Table 11). The adaptive schedule dominates the cosine schedule at *every* agreement level. The gains are largest in the lower-confidence tiers, where the agreement signal is most sensitive to how noise is annealed: the split-vote and moderate-majority tiers improve by +21.30 and +21.60 percentage points, respectively, and even the baseline single-path setting improves by +10.10 points. At the high-confidence tiers the schedules converge but the adaptive variant still leads (+7.25 points on unanimous answers, +2.50 on strong majorities). This indicates that front-loading exploration and rapidly tightening the noise yields better-calibrated trajectories than a smoothly decaying cosine schedule, particularly for harder questions where early-stage exploration matters most.

# D  Language Mixing

This appendix provides a representative example of the spontaneous language-mixing (code-switching) behavior discussed in Section 5. During our experiments, we observed that at certain noise levels the perturbed latent trajectories occasionally produced reasoning traces that switched between English and another language (most commonly Chinese), even though all prompts were English-only. Intriguingly, these code-switched traces sometimes arrived at the correct answer on problems where the model's standard monolingual reasoning failed. The example below illustrates one such case and is intended to make the phenomenon concrete; it is anecdotal rather than a controlled result, and we did not pursue it systematically (see Section 5 for discussion and Wang et al. (2025) for a systematic study of the underlying effect).

The example concerns a single GSM8K word problem (shown in the *Input Question* box, with reference answer 285). We compare two reasoning paths generated for the same question under noise injection. The first (*Response A*) reasons entirely in English and makes an arithmetic slip: when computing the popping rate for the final interval, it halves the second-interval count (60) rather than the third-interval count (120), propagating the error to an incorrect final answer of 247. The second (*Response B*) is a code-switched trace that carries out the same calculation but performs the disputed step correctly ($120 \times 0.5 = 60$), yielding the correct answer of 285. Because the code-switched response is written largely in Chinese, we provide an English translation immediately below it. Chinese text segments are highlighted in yellow, and the translation was provided by Hannah Cyberey.

We stress that this example is illustrative. It shows that language mixing *can* coincide with a recovered correct answer, not that it reliably does so. We lacked a principled mechanism to predict when code-switching would help or to induce it on demand, which is why we flag it as a direction for future work rather than a contribution of this paper.

---

### Input Question

**Question ID:** 1                                                          **Reference Answer:** 285

Gabriel is popping popcorn for a snack. As the skillet of kernels heats up, the kernels start popping faster. 30 pop in the first 20 seconds of cooking, then 2 times that amount in the next 20 seconds. The kernels increase to 4 times the initial popping rate in the next 20 seconds, but in the final 20 seconds, the popping slows down to half the rate as the past 20 seconds. After Gabriel takes the skillet off the heat, a quarter of the number of kernels that popped in the final 20 seconds of cooking also pop from the residual heat. How many pieces of popcorn does Gabriel have to eat?

---

### Response A: Monolingual (English)

Initial popping rate is 30 kernels in 20 seconds.
In the next 20 seconds, the popping rate is 2 times the initial rate:

$$2 \times 30 = 60 \text{ kernels}$$

In the next 20 seconds, the popping rate is 4 times the initial rate:

$$4 \times 30 = 120 \text{ kernels}$$

In the final 20 seconds, the popping rate is half of the previous 20 seconds:

$$\frac{1}{2} \times 60 = 30 \text{ kernels}$$

*← Error: Should be half of 120, not 60*
After taking the skillet off the heat, a quarter of the final 20 seconds' popping also pops from the residual heat:

$$\frac{1}{4} \times 30 = 7.5 \text{ kernels}$$

Adding all the popped kernels together:

$$30 + 60 + 120 + 30 + 7.5 = 247.5 \text{ kernels}$$

Since we can't have half a kernel, we'll round down to the nearest whole number:

**Final Answer: 247**  ✗ Incorrect

---

## Response B: Code-Switched (Chinese + English)

总共有几个爆米花?

首先，我们来计算每个阶段爆米花的产生数量。

1. 第一阶段 ：前 20 秒，30 个爆米花。

2. 第二阶段 ：接下来的 20 秒，爆米花数量是第一阶段的 2 倍，即 $30 \times 2 = 60$ 个。

3. 第三阶段 ：接下来的 20 秒，爆米花数量是第一阶段的 4 倍，即 $30 \times 4 = 120$ 个。

4. 第四阶段 ：最后 20 秒，爆米花数量是第三阶段的 0.5 倍，即 $120 \times 0.5 = 60$ 个。
   *Correct: Half of 120*

现在，我们计算在最后 20 秒的爆米花中，由于残余热量，额外爆开的
爆米花数量。这是最后 20 秒爆米花数量的四分之一，即 $60 \times \frac{1}{4} = 15$ 个。

最后，我们将所有阶段产生的爆米花数量加起来：

$$30 + 60 + 120 + 60 + 15 = 285$$

因此， Gabriel 有 285 个爆米花可以吃。所以，答案是：

**Final answer: 285**  ✓ Correct

---

## Translation of Code-Switched Response

*How many pieces of popcorn are there?*
*First, let's calculate the popping rate for each step:*

1. **First step**: *the first 20 seconds, 30 pieces of popcorn.*

2. **Second step**: *In the next 20 seconds, the number of popped kernels is 2 times the amount from the first step, i.e. $30 \times 2 = 60$ kernels.*

3. **Third step**: *In the next 20 seconds, the number of popped kernels is 4 times the amount from the first step, i.e. $30 \times 4 = 120$ kernels.*

4. **Fourth step**: *In the last 20 seconds, the number of popped kernels is 0.5 times the amount from the third step, i.e. $120 \times 0.5 = 60$ kernels.*

*Now, we calculate the number of popped kernels in the last 20 seconds' popping from the residual heat. This is a quarter of the final 20 seconds' popping, meaning that $60 \times \frac{1}{4} = 15$ kernels.*

*Finally, we sum up all the numbers of popped kernels from every step:*

$$30 + 60 + 120 + 60 + 15 = 285$$

*Thus, Gabriel has 285 pieces of popcorn that he can eat. Therefore, the answer is:*

**Final answer: 285**    ✓ Correct

*Figure: Comparison of model outputs demonstrating noise-induced code-switching behavior. At certain noise thresholds, the model exhibits language mixing (Chinese-English) which correlates with improved reasoning accuracy on this mathematical task. Yellow highlighting indicates Chinese text segments in the code-switched response. Translation provided by Hannah Cyberey.*

