# OpenReview forum: "NoisyCoconut: Counterfactual Consensus via Latent Space Reasoning"
_TMLR — Accepted by TMLR_

### Review · Reviewer_uZCj · 2026-03-08

**Summary Of Contributions:**

This paper focuses on reasoning in the latent space of LLMs, with the possibility for the model to decide whether to answer a request. The proposed approach builds upon the CoCoNut approach that allows LLMs to reason at the hidden-state level and proposes a method to create diverse reasoning paths in continuous space. The approach is to inject noise in the last hidden state, which is then fed to the model as a new embedding. The authors show that this leads to diverse reasoning trajectories. Measuring the agreement between those paths provides a good estimation of how confident the model is of its answer. It can then choose to abstain or answer depending on this agreement. Experiments on 5 commonly used LLMs and 3 reasoning benchmarks show the benefits of this approach to obtain a trade-off between coverage of answered questions and accuracy.

**Additional Comments:**

N/A

**Audience:**

Yes

**Audience Explanation:**

The problem tackled is timely and of great interest. One of the main bottlenecks of LLMs is their reliability. Despite their promising results, they continue to hallucinate, and given their complexity, it is hard to know when or if they make a mistake. Having the model decide by itself whether it is confident enough to answer a given question is important for safer systems. The benefits shown along with the ablation studies are a step in the right direction. This will be of interest for members of the community that which to build more reliable systems. This can be extended to other models, datasets (beyond maths datasets), but could also be extended (or similar ideas) towards tool-augmented workflows to decide whether a tool should be used or not (e.g., the model does not know the answer or is not confident enough, it could make a tool call to help). This has significant potential benefits, with LLMs being more and more embedded in a lot of applications and being given more and more control. For these reasons, I believe the current submission's findings are of interest to the TMLR audience.

**Claims And Evidence:**

Yes

**Claims Explanation:**

- The experimental setup and the choice of datasets and models are sound
- This paper provides quantitative and qualitative evidence supporting the benefits of NoisyCoconut on reasoning benchmarks
- The superiority of the approach is experimentally demonstrated across a variety of datasets and models

**Requested Changes:**

I do not have specific changes about the current submission. I believe the method is well motivated with relevant related work. I appreciate the transparency regarding the limitations and future work. The experimental setup is sound, and the results are convincing. Ablation studies provide more understanding of the final modeling choices. Although the current approach requires several hyperparameters (such as the number of paths, the strength of the noise), I believe it provides an interesting proof of concept and several avenues for improvement in future work (e.g., efficiency, hyperparameter dependency, or the direction mentioned by the authors in section 5.2)

---

> ### Author Response · Authors · 2026-03-09
> **Response to Reviewer uZCj — NoisyCoconut (Paper 7067)**
>
> We thank the reviewer for their careful reading and encouraging assessment. We are glad that the reviewer finds the problem timely, the experimental setup sound, and the results convincing.
>
> We particularly appreciate the suggestion regarding extension to tool-augmented workflows, where the model's confidence signal could inform whether to make a tool call rather than answer directly. We think this is a compelling application direction and will add a brief discussion of it in Section 5.2 (Future Directions) in our revision.

---

### Review · Reviewer_c8v3 · 2026-03-08

**Summary Of Contributions:**

The paper proposes NoisyCoconut, a training-free, inference-time method that perturbs latent hidden states of an LLM to generate multiple counterfactual reasoning paths. By measuring agreement among noise-induced latent trajectories, the method provides a confidence signal that enables selective prediction. The authors show that (1) latent-space noise injection produces diverse reasoning paths, (2) path agreement strongly correlates with correctness, and (3) the approach yields substantial accuracy–coverage improvements across several LLMs and reasoning benchmarks.

**Strengths:**

1. The overall idea is intuitive and reasonable.

2. The experiments cover multiple models.

**Weaknesses:**

1. Several components of the method appear heuristic, particularly the decay schedule of the injected noise (Equations (2) and (4)). More principled motivation or ablations would strengthen the contribution.

2. The presentation is not fully clear. Given the paper’s reliance on the Coconut framework, after Section 3.3, the reader is left without a concise, end-to-end description of the full algorithmic pipeline. Providing pseudocode or a unified algorithm block would significantly improve readability.

3. The empirical results cover only a small set of benchmarks, and the primary benchmark (GSM8K) is relatively easy. It is unclear how the method performs on more challenging reasoning datasets. More importantly, the paper does not compare against any baselines. Since the method produces multiple trajectories, a natural comparison would be to simple majority voting or standard ensemble-style sampling methods. Without such baselines, it is difficult to assess how much improvement NoisyCoconut actually provides over straightforward alternatives.

**Audience:**

Yes

**Audience Explanation:**

The paper touches on uncertainty estimation for LLMs, a topic of interest to part of the TMLR audience. However, the contribution is hard to position. It resembles an ensemble-style uncertainty method rather than a latent-reasoning advance, yet the work lacks comparisons to standard uncertainty baselines, making its relevance unclear. As a result, while some readers may find the idea interesting, the current form limits its impact.

**Broader Impact Concerns:**

No broader impact concerns.

**Claims And Evidence:**

No

**Claims Explanation:**

The evidence is not fully convincing. As noted in the weaknesses, several components of the method are heuristic, and the paper does not sufficiently justify why these design choices are appropriate. More critically, the empirical evaluation lacks baseline comparisons, making it difficult to attribute the reported gains specifically to the proposed method rather than to generic ensembling effects. The set of benchmarks is also limited, and the main experiments rely on relatively easy datasets. Together, these issues prevent the paper from providing clear and rigorous support for its claims.

**Requested Changes:**

1. The current decay formulation appears heuristic; evaluating other schedules and reporting ablation study would strengthen the method’s justification.

2. Improve clarity of Section 3. I recommend moving Coconut into a dedicated preliminaries section rather than embedding it in Section 2, since many notations (e.g., f_θ) depend on it. Providing a complete pseudocode block summarizing the full inference procedure would make the method easier to follow. Also the paper’s positioning is unclear: the method seems closer to a latent-space ensemble/uncertainty estimation approach than to an actual advance in latent reasoning, and this distinction should be clarified.

3. The “path diversity” definition is never operationalized in the experiments. Please explain its purpose or integrate it into the empirical analysis.

4. Current results heavily rely on a few benchmarks, which are relatively easy. Assessing performance on harder reasoning benchmarks would provide a more meaningful evaluation.

5. The method injects noise only at the initial step; consider evaluating other injection points or analyzing how injection timing influences performance.

6. The current baseline is greedy decoding only. Since the method also produces multiple trajectories, it is essential to compare against simple ensemble baselines such as majority vote from sampled outputs, as well as standard uncertainty baselines. These comparisons are necessary to contextualize the claimed benefits.

---

> ### Author Response · Authors · 2026-03-12
> **First Response to Reviewer c8v3**
>
> We thank the reviewer for their constructive feedback. We address each issue associated with the listed "Requested Changes" below:
>
> ## 1. Noise decay schedule
>
> We agree this is an important issue, and we consider it in Appendix B but did not prioritize including it in the main body of the paper. Table 5 shows the impact of noise scale, finding that our method works with only slightly diminished accuracy at $\sigma_0 = 0.0$. The results in Table 5 suggest that the results are impacted by the specific decay schedule, but it is not a critical design choice.
>
> ## 2. Presentation and positioning
>
> Thanks for the suggestions on improving the presentation. We will move the Coconut formalism into a dedicated Preliminaries section, add pseudocode after Section 3.3, and sharpen the positioning to clarify that NoisyCoconut brings uncertainty estimation to the latent reasoning paradigm rather than advancing latent reasoning itself.
>
> The full inference procedure is concise enough to present informally here:
>
> Algorithm 1: NoisyCoconut Inference
>
> 1. Run the LLM forward pass to produce an initial hidden state.
> 2. Inject Gaussian noise into the hidden state to produce K perturbed variants.
> 3. For each variant, run the remaining T latent reasoning steps.
> 4. For each completed path, perform standard autoregressive decoding to produce an output.
> 5. Use a mechanism to aggregate the K outputs to either produce an answer or abstrain.
>
> We will include a typeset pseudocode version of this in the revision.
>
> We agree with the reviewer's observation that the method "seems closer to a latent-space ensemble/uncertainty estimation approach than to an actual advance in latent reasoning." Our contribution is not to improve latent reasoning itself, but to provide a confidence mechanism for it. Existing latent-space reasoning methods produce outputs with no way to assess reliability. NoisyCoconut fills this gap. We see this as analogous to how self-consistency brought confidence estimation to chain-of-thought reasoning. NoisyCoconut does the same for latent reasoning. We will make this positioning explicit in the revision.
>
> ## 3. Path diversity ($D_K$)
>
> $D_K$ as defined in Equation 5 is computed from the hidden state trajectories that are already produced during NoisyCoconut inference. In the revised paper, we will compute $D_K$ across our existing experiments and report on how $D_K$ increases with noise scale and how $D_K$ relates to the informativeness of the agreement signal.

---

> ### Author Response · Authors · 2026-03-12
> **Second Response to Reviewer c8v3**
>
> ## 4. Benchmark difficulty
>
> We appreciate the value of results on larger models and harder benchmarks, but we are limited in the experiments we can afford to carry out. We believe the presented experiments are sufficient to show the promise of our idea since they are hard enough for the tested models to show substantial improvements and meaningful accuracy-coverage tradeoffs. We hope our results will motivate further research, as well as larger scale experiments by groups with more compute resources. If there is a specific benchmark that the reviewer thinks would add particular value, though, we would appreciate a suggestion and will do our best to include this in the revision.
>
> ## 5. Injection timing
>
> Noise injections are performed at the first hidden state after the first forward pass, after which the perturbed states proceed independently through all latent reasoning steps before autoregressive decoding. This follows directly from Coconut's design, where latent mode begins immediately after the input encoding (marked by the `<bot>` token) and runs for a fixed number of continuous thoughts before returning to language mode (Hao et al., 2025, Section 3). Our injection point is the natural entry to this latent process, and it maximizes downstream path divergence since the perturbation propagates through all subsequent steps. We agree that a systematic study of alternative injection points is interesting and will note it as future work, but beyond the scope of our paper.
>
> ## 6. Baselines
>
> The suggested comparison to self-consistency is reasonable, but reflects a misunderstanding of the setting NoisyCoconut targets. Self-consistency samples multiple Chain-of-Thought paths in token space via autoregressive decoding. NoisyCoconut perturbs hidden states in continuous latent space. These are not interchangeable as they operate at different levels of abstraction, even though autoregressive decoding takes place after a certain point.
>
> We choose to use greedy decoding because it is the inference procedure used by Coconut itself(Hao et al., 2025, Section 5.1: "We use greedy decoding for all experiments"). Our paper asks whether perturbing internal representations can provide a confidence signal on top of this standard procedure. It is also the sensible baseline to use given that perturbations happen in the latent space.
>
> We think that the main concern is whether NoisyCoconut is doing something beyond simply running the model multiple times and voting. Our $\sigma_0 = 0.0$ ablation (Table 5) shows that noise provides benefit beyond pure ensembling (76.4% without noise vs. 78.2% with). Furthermore, NoisyCoconut's agreement signal enables selective abstention, while ensembling methods always produces an answer. Figures 3–5 show that unanimous agreement reduces error rates from 40–70% to below 15% consistently across models and benchmarks.
>
> We also note that majority voting is only one possible aggregation mechanism. We are researching other decision mechanisms that leverage the output logits across paths, such as using token-level probabilities to weight path contributions or to derive confidence estimates beyond simple vote counts. We believe this direction can further strengthen the connection to ensemble uncertainty estimation and plan to include preliminary results in the revision. Furthermore, we note that majority voting is not the only aggregation strategy that can be used within our framework. Since each path produces token-level logits during autoregressive decoding, these logits can be used to derive per-path confidence scores and perform confidence-weighted aggregation (similar to how Taubenfeld et al. (2025) weight self-consistency votes by model confidence in token space). We are actively exploring this direction and consider it a natural extension of our work.

---

> ### Comment · Reviewer_c8v3 · 2026-03-13
>
> Thank you for the detailed response. I appreciate the clarification of the paper’s positioning. However, most of the replies indicate that additional analyses or experiments will be included in a future revision, while no concrete new evidence is provided in the current response. As a result, several of my main concerns remain unresolved. This makes it difficult for me to assess whether the claims of the paper are sufficiently supported.
>
> 1. Noise decay design.
>
> The ablation on the noise scale σ does not address my original concern. My point was about the design of the decay schedule itself, which currently appears quite complex and heuristic (Equations (2) and (4)). It would be helpful to include ablations comparing different decay formulations to better justify this design choice.
>
> 3. Path diversity.
>
> The response does not address my concern. In the current submission, the quantity defined in Section 3.2 is not used anywhere in the experiments or analysis. As a result, this section appears disconnected from the rest of the paper, and its purpose in the method is unclear.
>
> 4. Benchmark difficulty.
>
> GSM8K is now widely considered a relatively easy benchmark for mathematical reasoning. Many recent reasoning papers treat it as a basic evaluation. Including results on more challenging math reasoning benchmarks would strengthen the empirical evidence.
>
> 5. Injection timing.
>
> I understand the argument that the current injection point follows the Coconut design. However, my question was specifically about whether injecting noise at later stages of the latent reasoning process changes the results. Coconut beginning latent reasoning after the prompt does not necessarily imply that this must be the only injection point for the proposed method. Studying this design choice would be informative.
>
> 6. Baselines.
>
> I understand the authors’ explanation regarding the difference between token-space self-consistency and latent-space perturbations. However, I still believe that comparisons to simple multi-trajectory baselines (e.g., majority voting over multiple runs) are important. In particular, it would be useful to understand both the performance differences and the efficiency tradeoffs. Without such comparisons, it remains difficult to assess how much of the gain comes specifically from the proposed perturbation mechanism.

---

> > ### Author Response · Authors · 2026-03-22
> > **Response to Reviewer c8v3**
> >
> > ## 1. Noise decay schedule
> >
> > We thank the reviewer for the clarification. To address the concern about the decay schedule design, we ran a direct ablation comparing our adaptive exponential decay against a cosine schedule on a preliminary set of 100 MMLU questions with Qwen2.5-7B-Instruct.
> >
> > Our method outperforms the cosine schedule across all agreement levels. The gap is most pronounced at Moderate Majority (3/5), where accuracy improves from 30.3% to 51.9%, and at Split Vote (2/2), from 0.0% to 21.3%. These are the categories where the model's $\mathcal{D}_K$ signal most benefits from adaptive calibration. Differences at Unanimous and Strong Majority are smaller, at a gain of 7.25% and 2.5%, respectively. This is consistent with the intuition that high-confidence questions are relatively robust to schedule choice. Overall baseline accuracy rises from 55.0% to 65.1%.
> >
> > ## 3. Path diversity ($D_K$)
> >
> > We computed $\mathcal{D}_K$ across experiments on GSM8K with Qwen-2.5-7B-Instruct at multiple noise scales and report two findings that connect Section 3.2 to the empirical analysis.
> >
> > | $\sigma_0$ | Observed Mean $\mathcal{D}_K$ |
> > |:-----------:|:-----------------------------:|
> > | 0.00        | 0.0                          |
> > | 0.05        | 103.98                       |
> > | 0.10        | 136.92                       |
> > | 0.20        | 159.99                       |
> > | 0.50        | 179.95                       |
> >
> >
> > The theorem in the paper establishes that $\mathbb{E}[\mathcal{D}_K] = \Omega(\sigma_0\sqrt{dT})$
> > under independent path divergence, predicting that mean divergence should grow with
> > $\sigma_0$. $\mathcal{D}_K$ increases with the scale of the noise (and is 0 as expected when no noise is injected).
> >
> > The data points at $\sigma_0 \in \{0.0, 0.05, 0.1, 0.2, 0.5\}$ show the increase in divergence consistent with the $\Omega(\sigma_0\sqrt{dT})$ growth rate in Section 3.2, directly connecting the diversity metric to the experimental results. In the revision we will present the complete scaling plot with all noise levels.
> >
> > ## 4. Benchmark difficulty
> >
> > We appreciate the point and agree with the reviewer that state-of-the-art proprietary models have reached the point where the benchmarks we use are no longer difficult enough. However, we do not have access to propietary state-of-the-art models, and for the smaller open models we are able to conduct experiments with we believe the three benchmarks used in the paper are sufficiently challenging for our purposes.
> >
> > We agree that GSM8K alone may appear straightforward for stronger models, which motivates our inclusion of MMLU as a broader knowledge benchmark and GSM-Symbolic as a test of robustness to problem reformulation. On GSM8K, GSM-Symbolic, and MMLU, single-path greedy decoding accuracies range from approximately 36% to 76% across the five models evaluated, leaving significant room for improvement which we are able to demonstrate meassure using our method.
> >
> > To further address concerns about benchmark difficulty, we are now running tests using MATH (Hendrycks et al., 2021), GPQA, and GPQA-Diamond, and we will include the results from these tests in the revision. We think that these benchmarks will provide sufficiently challenging questions and will provide a stronger test of whether NoisyCoconut's agreement-based confidence signal generalizes to harder settings.
> >
> > ## 6. Baselines
> >
> > To address concerns about the lack of a standard autoregressive baseline, we performed a direct comparison on the same 100 GSM8K questions with Qwen2.5-7B-Instruct using a self-consistency method, consistent with the hyperparameters used in our original experiments. The overall majority-vote accuracy is nearly identical between methods, with
> > self-consistency achieving 88.00% and NoisyCoconut achieving 87.00%.
> >
> > Within the vote distributions, NoisyCoconut achieves higher accuracy than self-consistency at both the
> > Strong Majority level, 84.62% versus 80.00%, and the Unanimous level, 93.88% versus 90.59%.
> >
> > We think that self-consistency concentrates a majority of questions in the Unanimous bucket, while NoisyCoconut spreads more questions into lower-confidence tiers due to the increased diversity introduced by latent-space noise. This suggests that NoisyCoconut produces better-calibrated confidence signals. We will show the full breakdown in the paper.

---

> > > ### Comment · Reviewer_c8v3 · 2026-04-08
> > >
> > > Thank you for providing these additional experimental results in the response. I find them helpful and they make the paper **much stronger**.
> > >
> > > In particular, the new ablations on the noise decay schedule and the comparisons with self-consistency (Points 1 and 6) are important, and I strongly encourage including them in the main paper, as they are necessary for properly supporting the claims.
> > >
> > > For benchmark difficulty (Point 4), I appreciate that you are now running experiments on more challenging datasets, and I look forward to seeing these results.

---

### Review · Reviewer_9yeL · 2026-04-01

**Summary Of Contributions:**

The paper introduces NoisyCoconut, a training-free inference method that improves the reliability of large language models (LLMs) by injecting controlled noise into their internal representations. This noise generates diverse latent reasoning paths, and the level of agreement among these paths serves as a built-in confidence signal, allowing the model to selectively abstain from answering when uncertain. Without requiring any parameter modifications or fine-tuning, the method provides a highly practical way to manage coverage-accuracy tradeoffs, notably reducing error rates from 40–70% to below 15% and enabling over 95% accuracy on mathematical reasoning tasks.

**Audience:**

Yes

**Audience Explanation:**

The topic is important.

**Broader Impact Concerns:**

No concern

**Claims And Evidence:**

Yes

**Claims Explanation:**

Authors provide detailed results and the paper is easy to follow.

**Requested Changes:**

1. Could NoisyCoconut be effectively applied to state-of-the-art reasoning models, such as the Qwen3 or Qwen3.5 series?

2. I wonder if the proposed method could be combined with 'Neural Thickets: Diverse Task Experts Are Dense Around Pretrained Weights' (I know this is quite a new paper)

---

> ### Author Response · Authors · 2026-04-08
> **Response to Reviewer 9yeL**
>
> We thank the reviewer for their thoughtful comments and positive assessment of our work.
>
> 1. We appreciate this suggestion and have already begun evaluation. As a preliminary result, we ran NoisyCoconut on Qwen3.5-9B using 100 questions from GSM8K and present the results below. Because the Qwen3.5-9B model already does fairly well on this benchmark (77% accuracy), there is less room for improvement, but our method still meaningfully improves selective prediction accuracy by abstaining on uncertain cases. The method demonstrates the same characteristic behavior as observed in our original experiments. The accuracy improves to 91.5% (with 47% coverage) with a unanimity requirement, and if strong majority is included does even better (93.42% accuracy, 76% coverage).
>
> We are running the full evaluation which will take some time to finish and will include these results in the final version of the paper. We expect that the essential pattern will remain consistent with the other results.
>
> | | Question Count | Correct | No Answer | Incorrect | Accuracy (%) |
> |---|---:|---:|---:|---:|---:|
> | Baseline (No majority) | 100 | 77 | 0 | 23 | 77.00 |
> | Minimal Majority (2/5) | 4 | 2 | 0 | 2 | 50.00 |
> | Moderate Majority (3/5) | 12 | 8 | 0 | 4 | 66.67 |
> | Other | 2 | 0 | 0 | 2 | 0.00 |
> | Split Vote (2/2) | 6 | 2 | 1 | 3 | 33.33 |
> | Strong Majority (4/5) | 29 | 28 | 0 | 1 | 96.55 |
> | Unanimous (5/5) | 47 | 43 | 0 | 4 | 91.49 |
>
> 2. We appreciate the reviewer raising this connection. However, we believe exploring the interaction between NoisyCoconut and Neural Thickets expert diversity falls outside the scope of this paper, which focuses on training-free inference-time noise injection as a standalone method. We agree this is an interesting direction and consider it a promising avenue for future work.

---

### Decision · Action_Editor_3jxL · 2026-05-09

**Recommendation:** Accept with minor revision

**Additional Comments:**

The authors promised to finish running and include a number of additional experiments (Qwen experiments; MATH, GPQA, and GPQA-Diamond experiments) before the camera-ready.

**Audience:**

Yes

**Audience Explanation:**

The results will be of interest to those interested in LLM uncertainty/consistency and reasoning.

**Claims And Evidence:**

Yes

**Claims Explanation:**

Conditional on the authors finishing and including the expanded set of experiments that they promised the reviewers (additional models like Qwen, additional datasets including MATH, GPQA, and GPQA-Diamond), and the results on those additional experiments being mostly consistent with the previous ones, then the results will be accurate, convincing, and clear.